# Exploring the Limits of Large Scale Pre-training

**Samira Abnar, Mostafa Dehghani, Behnam Neyshabur, and Hanie Sedghi**
Google Research
{samiraabnar,dehghani,neyshabur,hsedghi}@google.com

## Abstract

Recent developments in large-scale machine learning suggest that by scaling up data, model size and training time properly, one might observe that improvements in pre-training would transfer favorably to most downstream tasks. In this work we systematically study this phenomena and establish that, as we increase the upstream accuracy, performance of downstream tasks *saturates*. In particular, we investigate more than $4800$ experiments on Vision Transformers, MLP-Mixers and ResNets with number of parameters ranging from ten million to ten billion, trained on the largest scale of available image data (JFT, ImageNet21K) and evaluated on more than $20$ downstream image recognition tasks. We propose a model for downstream performance that reflects the saturation phenomena and captures the nonlinear relationship in performance of upstream and downstream tasks. Delving deeper to understand the reasons that give rise to these phenomena, we show that the observed saturation behavior is closely related to the way that representations evolve through the layers of the models. We showcase an even more extreme scenario where performance on upstream and downstream are at odds with each other. That is, in order to have a better downstream performance, we need to hurt upstream accuracy.

## 1 Introduction

Recent impressive progress on transfer and few-shot learning (Brown et al., 2020; Goyal et al., 2021; Kolesnikov et al., 2019; Pham et al., 2020; Dosovitskiy et al., 2020; Dumoulin et al., 2021; Radford et al., 2021) suggests an emerging direction that scaling up models and training them on a huge corpus of data is the main obstacle towards better performance on downstream tasks with less or no data. These developments implicitly encourage two consistent views: 1) scaling up the model and data size improves the performance significantly; 2) the performance improvement transfers to downstream tasks in a desirable way. In a more focused empirical study in support of the first view, Kaplan et al. (2020) show that scaling up the model size, data, and compute appropriately in the language modeling task results in a non-saturating return in performance. Bello et al. (2021),Tan & Le (2019) show that favorable scaling can be achieved in image recognition tasks as well. The second view has also been a subject of recent focused studies. Hernandez et al. (2021) show that favorable scaling laws similar to that of (Kaplan et al., 2020; Tay et al., 2021b) hold in transfer and few-shot settings in NLP tasks. In perhaps closest prior work to ours, Kornblith et al. (2019) observe a linear relationship[1] between the performances on ImageNet (Russakovsky et al., 2015) and downstream image recognition tasks.

Adopting the above views has major implications moving forward. These views suggest that spending compute and research effort on improving the performance on one massive corpus would pay off because that would enable us to solve many downstream tasks almost for free. It also means while improving our upstream performance, we do not need to be worried about downstream tasks as their improvement is predictable based on a linear trend. While the aforementioned studies provide a compelling story, they suffer from a major shortcoming: due to compute limitations, performance for different choices of hyper-parameter values are not reported. Scaling plots seem more favorable

---

[1]The linear relationship in (Kornblith et al., 2019) is achieved after proper logit scaling of accuracy values. We show that with logit or linear scaling, the relationship is not linear.

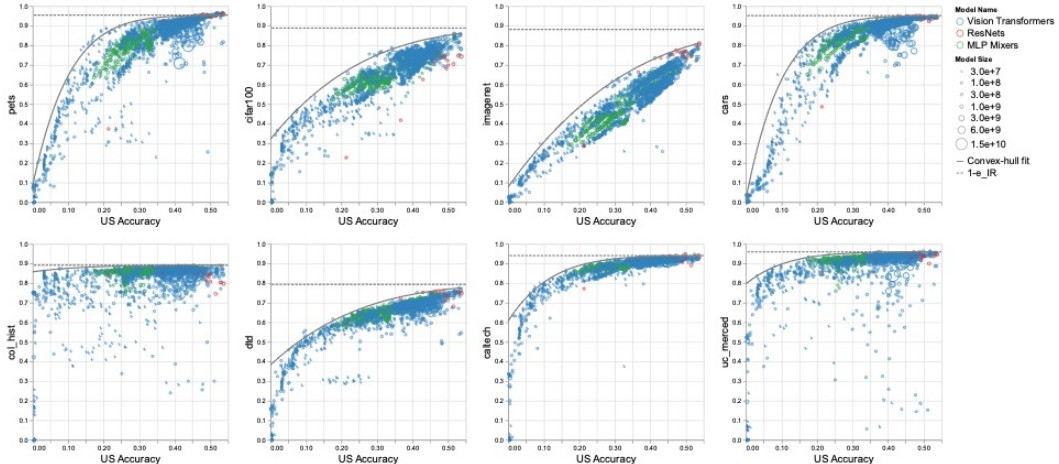

Figure 1: The performance of different downstream (DS) tasks vs of upstream (US) based on more than 1500 different Vision Transformers, 1400 MLP-Mixers and 16 best performing ResNets (Although the number of ResNet samples are small, this does not hurt our investigations. See Appendix G.1 for details), with different configurations. The models are pre-trained on JFT and evaluated in few-shot settings (25 shots). Figure F.1 in Appendix F shows the same plot but with more than 4800 experiments including two different upstream tasks of JFT, ImageNet21K and 1, 25 shots. We consider the convex hull of the points as well since it captures the performance of a randomized classifier made by choosing these models with different probabilities. *As the upstream performance improves, the downstream performance starts to saturate*. Even if US accuracy reaches 100% accuracy, the DS accuracy will not reach the 100% accuracy and saturates at a lower value. We observe a non-linear relationship between upstream and downstream accuracy and model the relationship with a power law function to predict the DS performance given the US performance. The horizontal line is the predicted downstream accuracy if upstream accuracy reaches 100%. We investigate DS-vs-US plots instead of the usual DS-vs-scale plots to capture the effect of hyper-parameter choices and to account for the fact that the scaling impacts DS performance through US performance. Figure F.2 depicts the same plot with log scaling of accuracies as done in many related works. Figure F.3 depicts the same plot when upstream is ImageNet21K.

if the hyper-parameter chosen for each scale is fixed or determined by a simple scaling function. Moreover, often the goal is improving state-of-the-art results, hence naturally most of the efforts in hyper-parameter selection is focused on higher scales, which significantly biases the scaling plots. However, when studying scaling, we are concerned about the best downstream performance of models given all possible values for the hyper-parameters. Additionally, most scaling studies report the behavior within a limited range, and simply extrapolating that scaling without further understanding of the dynamics of scaling can be detrimental as there is no reason, a priori, for the scaling to hold outside of the studied range.

In this paper, we systematically investigate the transferability of improvements on a large-scale upstream task to a wide range of downstream tasks in both few-shot and transfer learning scenarios. To address the above shortcomings, part of our work is a meta-study of more than 4800 Vision Transformer (Dosovitskiy et al., 2020), MLP-Mixer (Tolstikhin et al., 2021) and ResNet (Dosovitskiy et al., 2020) models. The models are pre-trained on either JFT (Sun et al., 2017) with 303M images and 18K classes or ImageNet21K (Deng et al., 2009) with 14M images and 21K classes and evaluated on a variety of downstream datasets for few-shot and transfer learning settings. Our 25 downstream tasks cover a wide range of standard datasets that are included in benchmarks like VTAB (Zhai et al., 2019), MetaDataset (Triantafillou et al., 2019), Wilds (Koh et al., 2020) and medical imaging.

We provide strong empirical evidence that scaling (and hyper-parameter tuning) does not lead to a one-model-fits-all solution. There are still many unresolved challenges remaining and at the center is the problem of data diversity for downstream tasks. We provide the first large scale and systematic investigation of this phenomena and discuss the reasons behind it. In Figure 1, we present downstream (DS) vs upstream (US) performance plot on variety of models and downstream tasks. We observe that, as we increase US accuracy, for most cases DS accuracy *saturates* to a value considerably below 100%. Also, saturating behavior is not an exception but rather the common trend and it is robust to the choice of number of shots and US tasks (see Figure F.1). We establish that this gap is not due to noise or any other factor that solely depends on DS task; rather, it depends on the relationship between US, DS tasks. Moreover, given a set of models with similar US accuracy, the best model for different DS tasks varies.

**Contributions**    Our main contributions in this paper are as follows:

- We establish through extensive study that as we improve the performance of the upstream (US) task either by scaling up or hyper-parameter and architectural choices, the performance of downstream (DS) tasks shows a *saturating* behaviour. In our experiments, several DS tasks reach full saturation within the studied range (Section 2).
- We demonstrate that given a set of models with similar US accuracy, the best model for a DS task $T_{DS_1}$ might have much worse performance on another DS task $T_{DS_2}$ compared to the best model for $T_{DS_2}$ (Figure 5).
- Given the scale of experiments, it is crucial for the proposed model to not be impacted by the density of the points in the DS-vs-US plot. We argue and demonstrate that fitting the power law to the convex hull of experiments would circumvent the effect of sampling biases on prediction of downstream accuracy and show the robustness of our model to sample size variations (Section 2.2).
- Having observed the nonlinear relationship between upstream and downstream accuracy, in order to predict downstream performance for a given upstream accuracy, we model their relationship with a power law curve and establish that it captures the behavior well even with small number of samples (Section 2.2).
- We study how scaling up model size, data size, and compute affects DS performance and show that these parameters impact DS performance mainly through the US performance (Section 3).
- We investigate reasons behind the DS performance saturation and show that this behavior can be captured by the usefulness of feature representation in higher layers of the pre-trained model (Section 4).
- We further explore the discrepancy between US and DS performances and show that for some choices of hyper-parameters, they might be at odds with each other. In particular, we showcase how the optimal hyper-parameters for the head used in pre-training (upstream task) are different for US and DS. We then uncover the reason behind this discrepancy (Appendix C, D).
- Finally, we show our observations are robust to several choices such as size of US data, common scalings of accuracy, number of shots, transfer vs few-shot setting and architecture (Appendix E).

**Related Work.**    The closest work to ours is that of Kornblith et al. (2019). They investigate the effect of ImageNet (Russakovsky et al., 2015) pre-training on image classification performance across 12 datasets for few-shot, transfer and random initialization scenarios. They show that performance on ImageNet translates linearly (in logit scaling) to performance on DS tasks. However, they do not consider extrapolation of the values. While both works investigate the effect of pre-training via various experiments, there are two main differences in our responses to the question of "better upstream performance transfer to better downstream performance?". First, we establish that clear "saturation" phenomena exists when looking into DS-vs-US performance. In Figure 1, we see there are various cases when comparing two models, A, B. Where model A has a much higher US accuracy but lower DS accuracy and these are not exceptions to a rule, rather the majority of cases. Essentially, for each DS-vs-US plot two points where one is on the right but lower than the other are instances of such a case. Second, we also establish that, for each DS task you can see best performing models scale with power law as in Equation 1 but for each architecture best performing models are different across DS tasks (Dehghani et al., 2021c) and this depends on training hyper-parameters, See Figure 5. In other words, when considering two DS tasks, $T_{DS_1}, T_{DS_2}$, we have numerous cases where model A has better performance on US and $T_{DS_1}$ but one cannot conclude better performance on $DS_2$. We suspect the difference in conclusion is due to the earlier work being limited in the range of accuracy values they consider. In addition to this difference in conclusions, we investigate the reasons behind this saturation behavior. Moreover, (in Appendix C) we consider cases where US and DS performance are at odds with each other, specifically, the scenarios where worse performance on US, leads to performance improvement on DS. Inspired by (Zhai et al., 2021) who noted that increasing head weight decay during pre-training leads to worse performance on US while improving DS performance; we investigate head hyper-parameters (both weight decay and learning rate) further and show that it can be explained by noting that these manipulations push the information stored in the head down to lower layers. Additional related work are covered in Appendix A.

## 1.1    EXPERIMENTAL SETUP

The analysis in this paper is based on a study on an exhaustive number of large-scale experiments on image recognition tasks, as well as a set of controlled experiments we conducted to ablate our

setup and deepen our understanding of the studied phenomena. We investigate more than $4800$ experiments with Vision Transformers, MLP-Mixers and ResNets with different configurations when pre-trained on a large amount of data in a supervised fashion and evaluated on several downstream image recognition tasks through few-shot learning and fine-tuning. For more details, see Appendix G.

We emphasize that the large set of experiments we investigate are not trained for the purpose of this paper, rather, we have aggregated models trained by different researchers for different purposes to perform a meta-study on them. This, in fact, positions this meta-study in a unique spot. First, it may not be feasible to run such a number of large-scale trials for the purpose of studying particular phenomena.Second, no implicit or explicit assumption was made in these experiments with respect to the type of analysis we conducted on them afterwards, hence minimizing the systematic biases of the analysis process in the findings. We note that, there might potentially be other biases. For example, researchers usually focus on hyper-parameter tuning to improve SOTA on a specific downstream task (usually ImageNet) and this may lead to not do a grid search on high dimensional space of all possible hyper-parameters and possibly affecting the plots. In Section 3, we investigate this and discuss that in this case the observed trend is similar to performing a grid search.

In the main body of the paper, we report the results over eight downstream tasks and provide results for more than 20 downstream tasks in Appendix F. Moreover, the plots corresponding to pre-training on JFT, ImageNet21K are in the main part and Appendix F, respectively.

## 2 THE DIMINISHING BENEFIT OF SCALING UP IN TRANSFER LEARNING

The prominent goal of transfer learning is to have a good performance on downstream tasks. The first question we address is how performance improvement on the upstream task impacts performance on different downstream tasks. We are interested in modeling this effect to facilitate prediction of downstream performance.

### 2.1 RECAP: RANDOMIZED CLASSIFIERS

Before diving deep into the DS-vs-US performance plots, we recap the concept of a randomized classifier since we will be using it extensively throughout this section.

Given two classifiers with US and DS accuracy $a_1 = (a_1^{US}, a_1^{DS})$, $a_2 = (a_2^{US}, a_2^{DS})$, one can make a *randomized classifier* by picking the output of the first classifier with probability $p_1$ and the output of the second classifier with probability $1 - p_1$ for each input independently. Then the accuracy of the randomized classifier will be $p_1 a_1 + (1 - p_1) a_2$. That is, the randomized classifier's accuracy is the convex combination of accuracy of the two classifiers. By sweeping the value of $p_1$, all the points on this convex combination path can be achieved. We can extend this notion to the case of more than two classifiers. As the next lemma states, the accuracy of such a randomized classifier would be a convex combination of accuracies of its endpoints.

**Lemma 2.1.** *Consider a group of models $\theta_j, j \in [N]$ that reach accuracy $a_j = (a_j^{US}, a_j^{DS}), j \in [N]$ on some pair of tasks (US,DS). Construct a randomized model $\tilde{\theta}$ as follows: for each input $x_i$, with probability $p_j$ pick model $\theta_j$ and output $\theta_j(x_i)$. Then the randomized model will demonstrate accuracy $\sum_{j=1}^{N} p_j a_j$.*

For proof, see Appendix B.

Therefore, all the points on the convex hull of DS-vs-US accuracy of the trained models are achievable and we have the aforementioned method to reach them. This leads to a randomized classifier that shows the accuracy equivalent to the convex hull of performances of trained classifiers at hand.

Based on the above discussions, in addition to the points corresponding to experiment results, we include the upper hull of the convex hull (representing the highest DS accuracy for every given US accuracy) of the model performances in our analysis. This provides us with a model of DS-vs-US relationship that is robust to density of the points in the plots. We discuss this further in Section 2.2.

## 2.2 SCALING LAWS FOR DOWNSTREAM ACCURACY

Figure 1 shows DS-vs-US performance for more than $3K$ experiments where different models are pre-trained on JFT and evaluated on a set of DS tasks in the few-shot setting (25 shots). Figure F.1 in Appendix F depicts a similar plot with all the $4800$ experiments (pre-trained on JFT or ImageNet21K, and for 1, 25 shots). As mentioned in Section 1.1, these models vary in terms of model size and shape, optimization method, compute and other hyper-parameters.

We are interested in predicting how the performance of a DS task will change if US performance improves by investigating the performance of existing models. To do so, we fit a curve to the DS-vs-US performance plot. We emphasize that our analysis differs from earlier works that analyze scaling law (Kaplan et al., 2020; Hernandez et al., 2021; Zhai et al., 2021) in that it analyzes DS accuracy vs US accuracy, instead of DS accuracy vs dataset size, model size or compute. Since for the most part performance improvement on US is achieved by scaling (dataset size, model size, compute), this approach indirectly captures the impact of scaling. We support this argument in Section 3.

When studying DS-vs-US choosing the right scaling is important. Kornblith et al. (2019) investigate DS-vs-US curve for models that are pre-trained on ImageNet and report a linear DS-vs-US performance when plotting the accuracies in the logit scaling. In Figure F.2, we depict the same experiments to that of Figure 1 but with logit scaling and we note a nonlinear relationship between DS and US accuracies. Recht et al. (2018; 2019) also use logit scaling for investigating relationship between DS and US performance. However, logit scaling shows a symmetric behavior around error 0.5, which is not natural for these problems. Therefore, we argue that log scaling which is used in scaling law literature is more appropriate. A linear relationship between US and DS performance in log scaling can be captured as follows:

$$e_{DS} = a(e_{US})^b.$$

Looking at Figure 1, we note that the behavior is not linear. Rather, the performance of DS task *saturates* at some point and that point varies for different DS tasks.

**Performance Saturation:** We define the saturation point inspired by the observations in Figure 1 and Figure F.1. Then, we mathematically model and investigate saturation value.

**Definition 2.2** (Saturation value). *Considering downstream vs upstream accuracy, for a downstream task $T_{DS}$ saturation value is defined as the value of downstream accuracy as upstream accuracy reaches* $1.0$.[2]

Considering Definition 2.2, performance saturation also means that there exists a US accuracy value, beyond which the performance improvement on DS is very small and hence it is not worth scaling up data size, compute or model size to improve US accuracy. Since the relationship is not linear, in order to predict DS performance, we need a function form to fit the plot. Inspired by recent work on scaling law (Kaplan et al., 2020; Hernandez et al., 2021), we propose the following function form:

$$e_{DS} = k(e_{US})^{\alpha} + e_{\text{IR}}, \tag{1}$$

where $e_{DS}, e_{US}$ refer to the error $(1-$ accuracy$)$ of downstream and upstream respectively, $k, \alpha$ are constants and $e_{\text{IR}}$ is the irreducible error. Irreducible error, $e_{\text{IR}}$, captures the value of DS error if US error reaches zero and hence acts similar to a bias term. $e_{\text{IR}}$ term captures the nonlinearity trend between US, DS accuracies. Meaning that if we plot Equation 1 in log scaling, the dependencies are linear only when $e_{\text{IR}}$ is zero.

We sketch the line corresponding to $1 - e_{\text{IR}}$ in DS-vs-US accuracy plots of Figure 1 and note that it is not close to $1.0$ for many DS tasks and better US performance does not transfer to better DS performance in higher US accuracies. We observe that unlike the common belief, the saturating behavior is not an exception, but typical among DS tasks.

**Effect of design choices on power law parameters:** As we can see in Figure F.1, different DS tasks have different saturating values and this value changes as US task changes. Moreover, $e_{\text{IR}}$ changes when we vary the number of shots. To depict this observations more clearly, we plot how different

---

[2]More precisely, saturation value is the value of DS accuracy when US accuracy reaches its Bayes error. This can be captured by replacing $e_{US}$ with $(e_{US} - e_{US-BayesError})$ in Equation 1. For simplicity and without loss of generality we do not account for upstream Bayes error in the discussions.

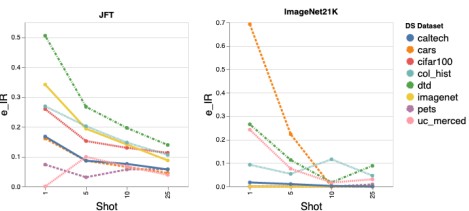

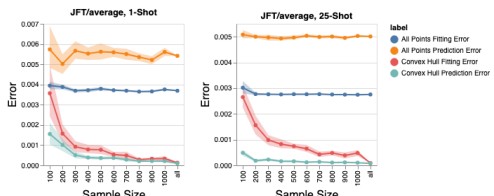

Figure 2: Effect of number of shots and DS/US task on $e_{\text{IR}}$ of the power law curves. We note that all of them impact $e_{\text{IR}}$. To see this effect for all power law parameters see Figures F.4, F.5.

Figure 3: The effect of sample size on power law curves. The curves are fitted to the convex hull of experiments as well as all data points from Figure F.1. Prediction error is the difference between power law prediction and observed value of the DS accuracy. Fitting error is the difference of power law values from the points that are used in fitting power law parameters. For more details see Appendix F.1.1

choices affect the parameters of the power law (Equation 1) in Figures 2, F.4, and F.5. It can be seen that the choice of US and DS task affects all parameters, while number of shots mostly impacts $k$ and $e_{\text{IR}}$. Specifically, increasing the number of shots results in lower $e_{\text{IR}}$. In short, there exists some functions $f_1(\cdot)$, $f_2(\cdot)$, and $f_3(\cdot)$ such that for a specific choice of model and training algorithm, we have

$$\alpha = f_1(T_{US}, T_{DS}, d), \quad k = f_2(T_{US}, T_{DS}, d), \quad e_{\text{IR}} = f_3(T_{US}, T_{DS}, d), \tag{2}$$

where $d$ refers to the number of shots in the few-shot setting.

To shed more light into this, we look into correlation of $k$, $\alpha$, and $e_{\text{IR}}$ with number of shots for different DS, US tasks in Table F.1. For all US and DS choices, $k$ and $e_{\text{IR}}$ correlate negatively with number of shots, while $\alpha$ is positively correlated with number of shots. However, correlation values change drastically for different choices of US, DS tasks. In addition, we look into the trend of each of these parameters as we increase the number of shots and present the likelihood of binary correlation in Table F.2. Both these tables capture similar phenomena.

**Irreducible error is not due to DS Bayes error:** We argue that irreducible error, $e_{\text{IR}}$, is not the Bayes error for DS task. Bayes error for a task refers to the error that is intrinsic to the definition of the task. More specifically, the Bayes error captures whether the classification labels are deterministic, i.e., there is a non-zero probability of a given instance belonging to more than one class. However, as can be seen in Figure F.4, for each DS task, $e_{\text{IR}}$ changes significantly by changing the number of shots and choice of US task. Therefore, $e_{\text{IR}}$ is not merely due to the Bayes error of DS task, but is also influenced by number of available samples from DS and the difference between US and DS tasks.

**Choice of data for fitting the power law:** As can be seen in Figure 1, there is a large variance in DS-vs-US performance across models. When considering the scaling law of the trained models, earlier works fit a scaling curve to all the existing points. We propose another option. To calculate the convex hull of all trained models and fit a scaling curve to the convex hull. The former essentially fits the scaling curve to average model performance. The latter has the advantage of fitting a scaling curve to the best performing models. The location of the points in the DS-vs-US plot significantly impact the average model and hence the power law prediction. Whereas, a convex hull of points is not affected by the locality of higher density points. A good performing model directly impacts the convex hull with no need to having many such samples. Therefore, we expect the average model to give an incomplete picture of the performance behavior. As we see below, fitting the convex hull is more robust to small sample size. Figure F.6 and F.7, depict the power law (Equation 1) curves corresponding to these two choices respectively. We plot the predictions from the power law curve on the higher US accuracies as well as the ground truth (prediction target) and observe that power law curve closely predicts the performance of DS. See Figure F.8, F.9 for 1 and 25 shot setting.

**Sample size sensitivity analysis:** We also investigate the robustness of this fit when we change the number of samples, in terms of error encountered when predicting the plot for higher US accuracies as well as the error in fitting the data. We use the points from higher US accuracies as held out data. Prediction error captures the difference between power law prediction and observed value of the DS accuracy. Fitting error captures the difference between power law values and the points that are used in calculating power law parameters. We plot fitting error and prediction error as the number of samples changes. Figures 3, F.10 summarize these errors when fitting the power law curve to the convex hull of DS-vs-US plot, and all data points. For more details, see Appendix F.1

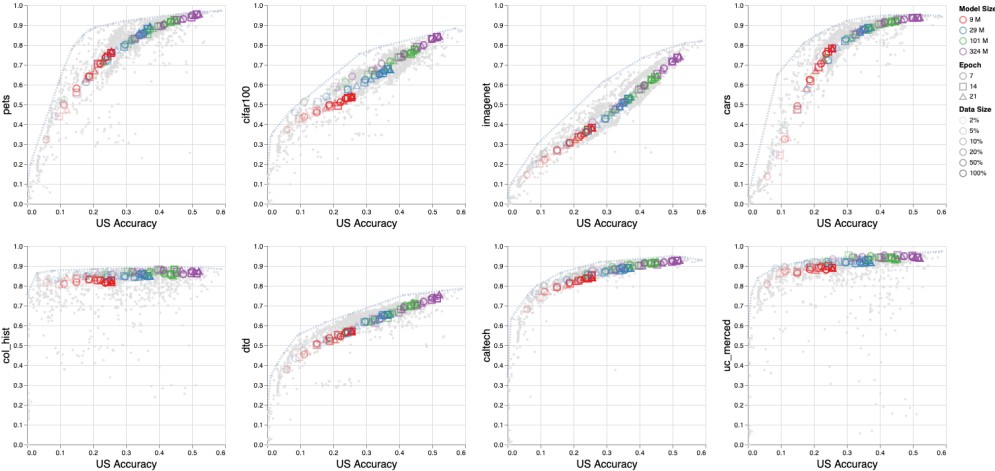

Figure 4: Controlled scale up experiments of the model size (number of parameters), data size (portion of the pre-trained data), and compute (number of epochs) on different downstream tasks and JFT as upstream. We observe similar trends to Figure 1. (1) As we increase US accuracy, DS performance saturates. (2) Increasing model size, US data size, and compute all lead to the same curve. (3) The variation from the curve is due to training hyper-parameters. (4) US accuracy has a stronger predictive power for DS accuracy compared to model size, US data size, and compute.

and Figures F.11-F.16. The prediction error is very small across all these choices, which means the proposed model will work well even with smaller number of DS-vs-US samples (trained models). As expected, the fitting error decreases by increasing the number of samples. Note that the prediction error is an order of magnitude lower if we fit the power law to the convex hull vs all samples.

## 3 EFFECT OF SCALE: A CLOSER LOOK

In this section, we perform a set of controlled experiments where we increase data size, model size, number of epochs. Figure 4 depicts how DS-vs-US accuracy changes as we increase US dataset size (from 2% to 100% of JFT), number of model parameters (ViT-Tiny, ViT-Small, Vit-Base, ViT-Large) and number of epochs (7, 14, and 21 epochs). See Figure F.17 for all 25 DS tasks. Since we are in the under-parametrized regime and far from saturating on JFT, the effect of increasing data size is equivalent to increasing training time and the performance on the US improves as we increase the training time (Nakkiran et al., 2020). In order to facilitate a comparison with earlier experiments, in Figure 4 we overlay the new points (shown in color) to that of Figure 1 (shown in grey).

**Similar trend:** We observe that the controlled experiments in Figure 4 show similar trends to that of Figure 1, Figure F.1, i.e., the DS-vs-US accuracy presents different trends for different DS tasks when scaling up dataset size, model size and number of epochs. For some DS tasks the performance saturates quicker, for instance, colorectal histology (col_hist) and UC-Merced land use dataset. Furthermore, similar to Figure 1, for some of the DS tasks, the benefit of scaling up diminishes gradually, for instance for Cars (Krause et al., 2013) or Caltech101 (Li et al., 2004).

**Grid search equivalence:** Effect of model size on improving both US, DS accuracy is more pronounced compared to data size and number of epochs. However, we note that if we keep any two of the three parameters fixed and increase the third one, the points reside on the same curve. In Figure 4 the effect of changing data size and number of epochs are on the same curve to that of changing the model size. Therefore, we can trust that even if we did a grid search on all these parameters, Figure 1 would still present the same picture.

**On the prediction power of US accuracy:** The above observations show that the effect of each of the three parameters (model size, US data size, compute) on DS accuracy is only through US accuracy. That means, conditioned on US accuracy, none of these three parameters provide extra information on DS accuracy. To depict this further, we evaluate the effectiveness of using US accuracy to predict DS accuracy as follows. Since we have a single value prediction, we consider our prediction based on fitting the power-law of Equation 1 and compare it to using average DS accuracy for predicting DS performance. Figure F.18 plots the error as well as the power law prediction plot for all DS tasks considered in this paper. In addition, we calculate the standard deviation of error (difference between

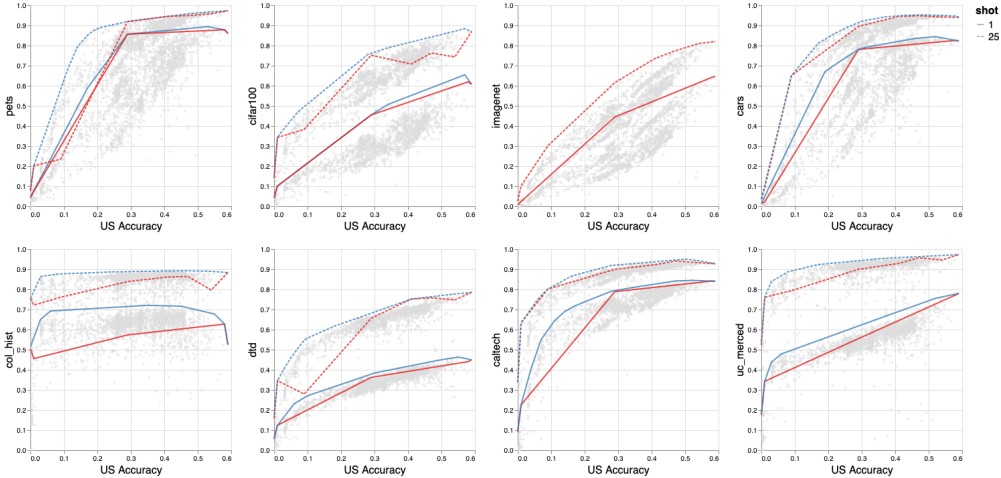

Figure 5: The overlay of the convex hull of ImageNet (Red) DS-vs-US plot on DS-vs-US plots of all DS tasks from Figure 1(Blue). US task is JFT. We observe that best performing ImageNet models perform very similar to best performing models in several DS tasks but not all of them. Moreover, as the US performance increases, the gap between best performing ImageNet models and best performing DS task models reduces significantly.

Equation 1's prediction of DS accuracy and the value of DS accuracy) and report in Table F.3. We note that the standard deviation of the error is much smaller than 1 (which is the standard deviation we would get if we used average as prediction value). This shows that US accuracy has a strong prediction power for DS accuracy and conditioned on US accuracy, there is not much left for the rest of parameters (model size, data size, compute) altogether to predict the DS accuracy. This further confirms our choice of parameter to rely on for predicting DS accuracy.

**On the role of hyper-parameters:** Moreover, contrary to (Hernandez et al., 2021), these three parameters (data size, model size, number of epochs) are not the only ones that impact the DS accuracy results. When we do controlled experiments on these three parameters, the points end up in the same curve. The variations observed in Figure 1 are due to different architecture and choices of training hyper-parameters and algorithms. The variations caused by the effect of hyper-parameters lead to the points not residing on the same curve in Figure 1. We observe a distance on the points corresponding to controlled experiments from the convex hull (best performing models). For example, for ImageNet, controlled experiments lead to a curve that is close to linear, however, this curve is in the middle of the curve from Figure 1, where in addition to scaling we change hyper-parameters and training details. We discuss the effect of hyper-parameters further in Appendix C.

## 4    INVESTIGATING DIFFERENT DS-VS-US TRENDS

In this section, we investigate the reason behind the saturation behavior in the DS-vs-US accuracy plots and address why saturation happens much earlier for some DS tasks compared to the others. First, we take a closer look at Figure 1 by overlaying convex hulls of different downstream tasks on top of each other. Specifically, we overlay the convex hull of ImageNet DS-vs-US plot on DS-vs-US plots of all DS tasks. Figure 5 and Figure F.19 show this for cases where US task is JFT and ImageNet21K respectively. We observe that (1) Best performing ImageNet models perform very similar to best performing models in several but not all DS tasks. (2) As the US performance increases, the gap between best performing ImageNet models and best performing DS task models reduces significantly. We also depict Spearman correlation between accuracies on different DS tasks, and between DS tasks and US task in Figure F.20 and F.21 respectively. Therefore, as the next step, we focus on capturing the difference between different DS tasks.

As discussed in (Yosinski et al., 2014; Neyshabur et al., 2020), lower layers capture lower level features that are more common across different dataset and tasks, whereas fine-grained features reside at top layers in the network. In addition, examples that are learned in higher layers are learned later in training with lower confidence and higher uncertainty (Baldock et al., 2021). Inspired by these observations, we measure the performance of few-shot classifiers when applied on top of representation from different layers of the pre-trained model. We look into the depth of the earliest layer that leads to the best performance for a given DS task and check whether this is a proxy of the difference between US and DS, and an indicator of how much the DS task will benefit from

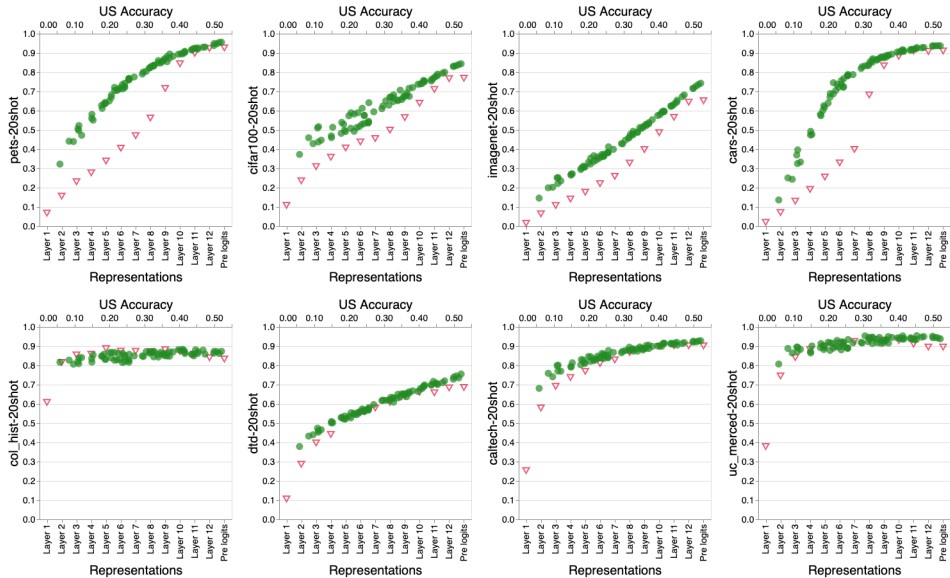

Figure 6: The effect of choosing representations from different layers on the downstream tasks performance overlay-ed with the effect of scaling (model, data, and compute) on downstream performance when upstream task is JFT. The red triangles are performance on downstream task when representation used in the few-shot learning is from different layers of the model. The green circles overlay the DS versus US performance of different experiments from Figure 4 on each task. Red triangles use the x-axis on the bottom and the green circles use the x-axis on the top. We note that for those DS tasks that are similar to US, such as ImageNet, the higher the layer the better performance on DS. On the contrary, for those DS tasks that saturate fast, such as UC-Merced and col_hist, the optimal layer is not the last one.

scaling up the compute or US data size. Figure 6, Figure F.22, F.23 present this result. We notice that, for DS tasks that are similar to US, such as ImageNet, the higher the representation layer the better performance on DS. On the contrary, for DS tasks that saturate fast, i.e., do not follow performance improvement on US, such as UC-Merced and col_hist, the optimal layer is not the last one. That is, choosing lower layers as the top layer and skipping the rest of the network leads to same or better performance on the DS.

Bringing the two discussions together, performance saturation on DS happens when the pre-trained network lacks the fine-grained features required to perform well on DS. Therefore, one can get similar performance on such DS task when cutting the top layers of the pre-trained model, as seen in Figure 6. Interestingly, as we see in Figure 6, and Figure F.23, when we overlay the DS-vs-US accuracy curves on DS accuracy-vs-layer depth curves, they follow almost exactly the same pattern, which means they are both good proxies for capturing the relation between US and DS tasks.

# 5 DISCUSSION AND CONCLUSION

We have investigated the role of scale in few-shot and transfer learning performance in image recognition and have established through extensive study that as we improve the performance of the upstream task either by scaling up or hyper-parameter and architectural choices, the performance of downstream tasks shows a *saturating* behaviour. In addition, have provided strong empirical evidence that contrary to common narrative, scaling does not lead to a one-model-fits-all solution. We have demonstrated the role of hyper-parameters and emphasize that one cannot hope to find one pre-trained checkpoint that performs well on all possible downstream tasks. We assert that we should refrain from focusing on the performance of only one downstream task, which usually ends up being close to the upstream task. Instead, we should make design choices that improve performance on a breadth of downstream tasks. Moreover, scaling has both monetary and environmental costs (Patterson et al., 2021; Dehghani et al., 2021a). We argue that, when investing in terms of scaling in terms of data, model parameters and compute, we should think of an additional axis which is *data diversity*.

The phenomena we described in the paper is not limited to the setting reported above. In Appendix E, we discuss that the observations are robust to several changes in the setting, namely, number of shots, few-shot vs transfer setting, scaling of plots and architecture.

Our paper focuses on the supervised image recognition task. Extending our investigation to unsupervised pre-training is also of interest. Exploring other modalities, e.g., natural language domain is the subject of future work.

## REPRODUCIBILITY STATEMENT

All the experiments conducted in this paper are based on Scenic library (Dehghani et al., 2021b). We have shared details on the customization done in the controlled experiments in Appendix 1.1. All other details, including data preprocessing steps as well as training and evaluation hyper-parameters, are kept fixed to their default. A proof for the theoretical discussion on randomized models (Section 2.1) is provided in Appendix B. We have also shared descriptions and references to all the downstream tasks and datasets we used for evaluation in Appendix G.3.

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

# Appendix

## A    ADDITIONAL RELATED WORK

Large scale transfer learning by pre-training on JFT (Kolesnikov et al., 2019; Dosovitskiy et al., 2020; Ryoo et al., 2021; Mustafa et al., 2021; Tay et al., 2021a; Puigcerver et al., 2020; Ngiam et al., 2018) or ImageNet21K (Dosovitskiy et al., 2020; Kolesnikov et al., 2019; Mustafa et al., 2021; Arnab et al., 2021; Likhosherstov et al., 2021; **?**; Puigcerver et al., 2020; Zhai et al., 2019) has been done extensively. Mensink et al. (2021) considers a two-step transfer chain, where the model is pre-trained on ImageNet, fine-tuned on the source task and then transferred to the target task. Then they look into the effect of different hyper-parameters on this transfer chain. They conclude that the effect of transfer learning vanishes as the target domain size increases. This is very different from the setting we consider, that is when the size of target domain is very small (few-shot setting).

Raghu et al. (2019) investigate the performance of models pre-trained on ImageNet when they are used to transfer to medical images. They conclude that the family of smaller lightweight convolutional networks performs comparably to standard ImageNet models, despite having significantly worse accuracy on ImageNet. Hence, ImageNet performance is not predictive of medical performance. Neyshabur et al. (2020) also studies transfer learning from models trained on ImageNet. They note that improved accuracy from pre-training can be achieved in less steps of fine-tuning than what is done in practice.

In the context of transfer learning, generally, model selection is a major step. Given a set of pre-trained models, and a downstream task, we need techniques or methods to help us predict the performance of the pre-trained models on the downstream task. There have been several efforts in this direction, including NCE[3], LEAPS[4], and most recently LogME[2]. The main focus of the aforementioned methods is to estimate the transfer performance of the models efficiently without actually having to fine-tune them on the given downstream task. In this paper, on the other hand, our aim is not to predict the transfer performance of a specific model. We are taking the big picture into account, and focusing on understanding the limits of scaling up, i.e., the dynamics of the scaling behaviour of DS performance with respect to US accuracy. The framework we propose and employ in this paper, can in general be used to answer the question of "How beneficial it would be for the downstream task if we improve the performance of the models on the upstream task by scaling up data, compute, model-size?". We show that having enough data points with a wide enough range of accuracies on the upstream task, one can fit a power-law curve that can be used to predict the best possible accuracy on the DS task with respect to any given us accuracy. Ideally, this would mean that we can predict the performance of models that do not yet exist. (We are predicting the best performance one could achieve on a DS task by pretraining on a given US task). Here we use few-shot accuracy as an indicator of the performance of the models on the downstream tasks. Methods such as LogME can be beneficial for such analysis, e.g., instead of tracking the few-shot performance of the models, one could look into other transferability metrics.

## B    PROOF OF LEMMA 2.1

*Proof.* Since $p_j$ are probability values we have $p_j \geq 0$ for all j, $\sum_{j=1}^{N} p_j = 1$. The proof follows the definition of accuracy and simple counting, as follows. Accuracy captures total number of correct predictions over total number of predictions. Let $\tilde{a}$ refer to accuracy of $\tilde{\theta}$, i.e., $\tilde{a} = (\tilde{a}^{US}, \tilde{a}^{DS})$, let

$n_{US}, n_{DS}$ refer to total number of predictions for upstream and downstream respectively. That is

$$\tilde{a}^{US} \overset{(1)}{=} \frac{1}{n_{US}} \sum_{i=1}^{n_{US}} I(\tilde{y}_i = y_i) \overset{(2)}{=} \frac{1}{n_{US}} \sum_{i=1}^{n_{US}} \sum_{j=1}^{N} p_j I(\theta_j(x_i) = y_i)$$

$$\overset{(3)}{=} \sum_{j=1}^{N} p_j \left[ \sum_{i=1}^{n_{US}} \frac{1}{n_{US}} I(\theta_j(x_i) = y_i) \right] \overset{(4)}{=} \sum_{j=1}^{N} p_j a_j^{US},$$

where (1), (4) are due to the definition of accuracy, (2) is achieved by the construction of the randomized classifier and (3) is due to commutative property of addition. Similarly,

$$\tilde{a}^{DS} = \frac{1}{n_{DS}} \sum_{i=1}^{n_{DS}} I(\tilde{y}_i = y_i) = \frac{1}{n_{DS}} \sum_{i=1}^{n_{DS}} \sum_{j=1}^{N} p_j I(\theta_j(x_i) = y_i)$$

$$= \sum_{j=1}^{N} p_j \left[ \sum_{i=1}^{n_{DS}} \frac{1}{n_{DS}} I(\theta_j(x_i) = y_i) \right] = \sum_{j=1}^{N} p_j a_j^{DS}.$$

Putting these two together gives us

$$\tilde{a} = (\tilde{a}^{US}, \tilde{a}^{DS}) = \left( \sum_{j=1}^{N} p_j a_j^{US}, \sum_{j=1}^{N} p_j a_j^{DS} \right) = \sum_{j=1}^{N} p_j a^j.$$

Note that, this is the definition of convex hull of $a^j$, $j \in [N]$.

$\square$

## C    DISCREPANCIES BETWEEN US AND DS PERFORMANCES: A CASE STUDY

In the last section, we observed that there exist cases where increase in the US accuracy does not translate to performance improvement in DS. In section 3, we investigated the hyper parameters that are relevant to scaling, i.e., number of training epochs, number of training examples, and model size.

In this section, we build on the observations in Section 3 on the role of training hyper-parameters in the DS-vs-US performance plot. Here, inspired by (Zhai et al., 2021) we focus on hyper-parameters related to the head (projection layer). Zhai et al. (2021) observed the impact of decoupling head weight decay on the performance of DS and US tasks. Specifically they noted that a higher head weight decay during pre-training leads to a worse performance on US, while improving DS performance. In this section, we take a closer look at the effect of the head (the projection layer).

We present cases where there are discrepancies between US and DS performances when we change head hyper-parameters. We investigate the phenomena observed in (Zhai et al., 2021) further and provide explanations on why this happens. Moreover, we show that one can observe a similar phenomena by decoupling and *decreasing* learning rate of the head during pre-training. In addition, for both head WD, LR, we conclude that the optimal value for each DS task depends on the DS task.

The experiments in this section, are aligned with discussion on Section 3 on the effect of hyper-parameters and show that when we consider a point on the DS-vs-US accuracy plot, changing the hyper-parameters may lead to moving in different directions toward the convex hull. It does not necessarily lead to a vertical improvement where you keep US accuracy fixed and increase DS accuracy. There can be even cases where improving DS accuracy comes at the expense of hurting US accuracy.

### C.1    EFFECT OF HEAD WEIGHT DECAY

Figure D.1 shows the performance for DS when we increase US head weight decay. In this experiment, the weight decay for the rest of the network is kept at 0.01. Figure F.24 in Appendix F.4 depicts the same plot for 25 downstream tasks. We observe that

- For US, increasing the head weight decay up to a threshold (optimum head WD) improves the performance on US and increasing it beyond that threshold leads to over-regularization and worse performance.

- The optimum value for head WD is different for US and different DS tasks. That is, there are cases where increasing WD on US head, results in deteriorating performance on US but improves performance for some DS tasks. Therefore, head weight decay is an important hyper-parameter and should be optimized for DS.

- The optimal head weight decay for different DS tasks can be very different, i.e., If we take different DS tasks into account when tuning the value for this hyper-parameter we will end up with different optimal values. This is illustrated in Figure D.3, F.25. In other words, there are cases where increasing or decreasing US head WD results in improved performance for a DS task and degraded performance for another DS task. Therefore, one cannot simply save a checkpoint of a model pre-trained on an upstream task and use it for all downstream tasks.

- The optimal weight decay for DS is usually higher than the optimal one for US, as also shown in (Zhai et al., 2021).

- The impact of increasing weight decay on the head is more prominent when the number of shots is lower. For example, we observe that the effect is more prominent on 1-shot performance on all DS datasets than on 20-shot performance.

- This phenomenon is robust to the number of training steps in US, i.e., increasing the number of training epoch does not change the trend. See Figure F.26 in Appendix F.4.

## C.2 EFFECT OF HEAD LEARNING RATE

Next, we look into the effect of decoupling head learning rate, i.e., changing the learning rate of the head relative to the learning rate of the rest of the network. In this experiment, the learning rate for the rest of the network is kept at 0.008. We notice similar patterns when decreasing head learning rate to that of increasing head weight decay. Figure F.27 shows the discrepancy between DS (Imagenet and Caltech) and US (JFT) when we change head learning rate. Considering the trend for all DS tasks, we note that impact of head learning rate on DS is different from impact on US. When we decrease head learning rate, for a number of DS tasks the performance remains the same or improves when US accuracy degrades.

We also look into optimal head learning rate for different DS tasks in Figure F.28 in Appendix F.4 and observe that it depends on the DS task and for an optimal performance, one needs to tune head learning rate for each DS task.

## D INVESTIGATING THE EFFECT OF HEAD HYPER-PARAMETERS

First, we investigate the $L_2$-norm of layers as a proxy of the amount of information stored in them, as we change the head WD. In this experiment, the WD for the rest of the network is kept at 0.01. We observe that as we increase the WD on the upstream task, the norm of the weights in the higher layers increases while it does not change much in lower layers. Figure D.2 shows the sum of the norm of all layers before the head as we increase head weight decay.[3] We observed a similar pattern in distance to initialization as we increase head WD, we do not see a change in lower layers, but the distance to initialization increases for higher layers as we increase head WD.

It has been widely discussed that a network's margin of error (also called prediction margin) can predict its generalization performance well (Neyshabur et al., 2017; Bartlett et al., 2017; Jiang et al., 2018). We refer to the margin for a single data point as the difference between the score of the correct label and the maximum score of other labels. We report average margin value over train data. The classical notion of margin refers to the scores at the head. More recently, Jiang et al. (2018) proposed the notion of margin at different layers which normalizes the score difference by the norm of gradient differences at that layer. Due to the correlation of margin to how well the model separates the data at each layer, in order to investigate this phenomena we look into how head margin and

---

[3]Figure F.29 in Appendix F.4 shows this trend for each layer separately.

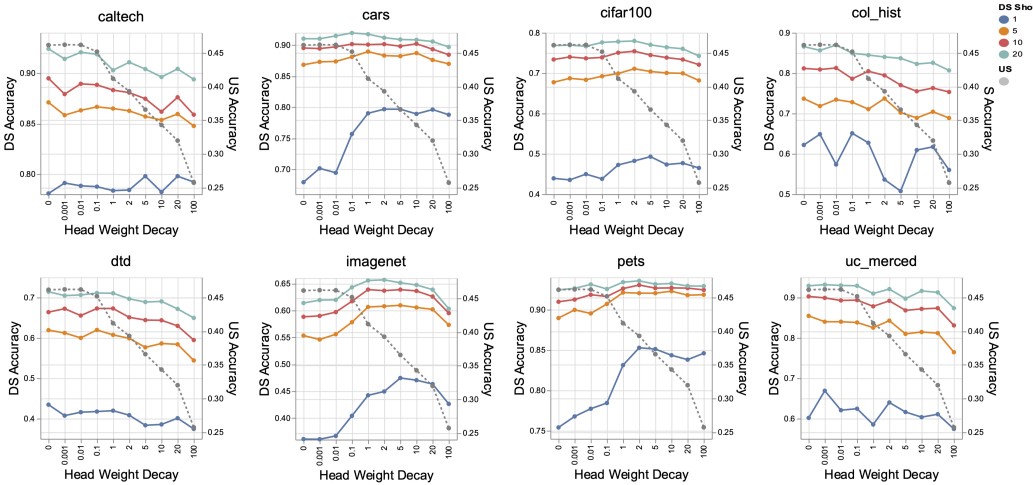

Figure D.1: The effect of increasing head weight decay in performance of upstream (JFT) versus performance of downstream (all shots). Note that not only the optimum value of head WD for upstream and downstream is different, but also the optimum value changes for different downstream tasks.

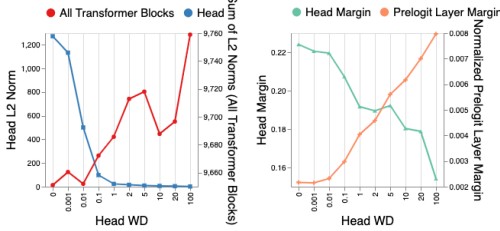

Figure D.2: Layer norm and layer margin for US as a function of head weight decay. As we increase the head weight decay, the sum of norms of all the layers up to the head, as well as the pre-logit layer margin increases, while the head's norm and margin decrease. Since these two metrics are correlated with amount of information stored in a layer, we conclude that increasing head weight decay pushes the information stored in the head to the layers below, similar to the effect of decreasing the head learning rate.

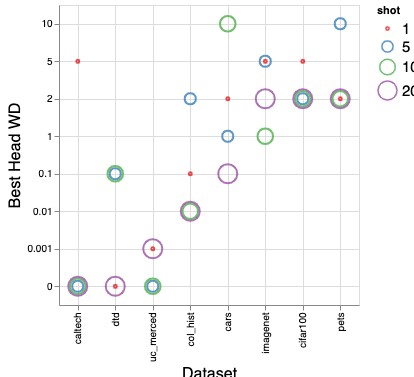

Figure D.3: Optimal head weight decay for each DS task for different number of shots. The optimum value is different for each DS task.

pre-logit (penultimate) layer margin changes as we increase head WD. We observe that as we increase the weight decay, the pre-logit layer margin increases, while the head layer margin decreases; See Figure D.2.

For US, although the head margin decreases with increasing head weight decay, which is also reflected in the performance drop on US (see Figure D.1), the margin for pre-logit improves. This shows that the information is being pushed down from head to pre-logit layer.

Since these two metrics are correlated with amount of information stored in a layer, the above two investigations suggest that as we increase head weight decay the information is pushed down to layers below the head. Moreover, these are still top layers in the network and the effect does not propagate nor affect early layers in the network.

Next, we look into the margin on DS datasets. We note that the margin (calculated on training data) trend completely reflects the accuracy trend on DS test data. Although this is expected in classical machine learning, it is still intriguing that we observe this pattern for a large-scale deep learning model where the margin has occasionally failed to capture generalization performance. We note that for datasets that saturate more slowly such as ImageNet, the margin increases as we increase head WD, and for datasets that saturate fast, such as Caltech101 and Cars, the margin does not change. See Figure F.30 in Appendix F.4 for DS margin plots.

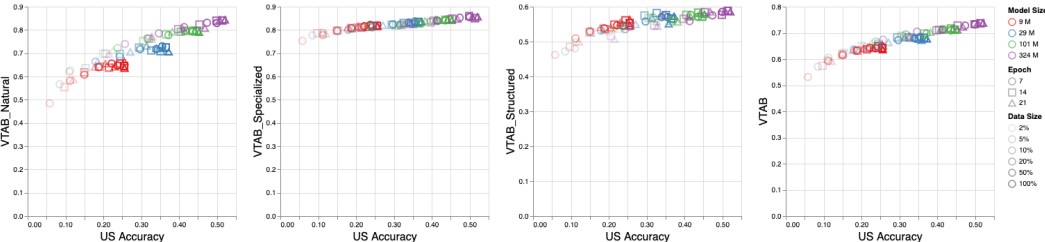

Figure E.1: Performance of models presented in Figure 4 in transfer learning setup on VTAB (Zhai et al., 2019) benchmark. VTAB defines a total of 19 tasks, grouped into three categories: (i) Natural, which contains *natural* images captured using standard cameras, *specialized*, which contains images of the world that captured through specialist equipment, and *Structured*, which contains tasks that designed to assess comprehension of the structure of a scene, mostly generated syntactically using simulated environments. In the transfer learning setup, we fine-tune all parameters of the pre-trained model as well as a randomly initialized head using all examples in the training set of the downstream task. Results for individual VTAB tasks are shown in Figure H.32. We observe a phenomena similar to few-shot setting. (1) Performance saturation happens in the transfer learning setting as well. (2) The effect of increasing model size, pre-training data size, compute lead to the same curve. (3) Strong power of US accuracy compared to model size, pre-training data size, compute in predicting DS accuracy.

We observe that as we decrease the head learning rate, the norm of the head decreases while the sum of the norm of other layers increases. A similar pattern is captured in US margin and norm plots when decreasing head learning rate as to increasing head weight decay (Figure F.31 in Appendix F.4). We note that the effects of these two interventions (increasing head WD, decreasing head LR) are similar. When we increase the head weight decay, as discussed above, we are pushing the information compressed in the network down to lower layers. On the other hand, when we decrease the head learning rate, we encourage lower layers to be more active and learn more. Both lead to the same impact.

Next, we look into the optimal WD as a function of the rank correlation between the performance of US and DS in Figure D.4. We calculate the rank correlations as follows. Given the list of model checkpoints, we make two rank lists based on US and DS performance and then calculate the correlation between the two lists. It can be observed in Figure D.4, that for a DS task optimal WD is high when we have a high correlation between performance on US and DS. The reason being when the correlation is high, one would want to move all the information that resides in the head to the lower layers and not lose any information. Since the head is removed for few-shot transfer, storing more information in the rest of the network leads to better performance in the DS. But when US and DS are different and hence uncorrelated, we do not need a high WD as

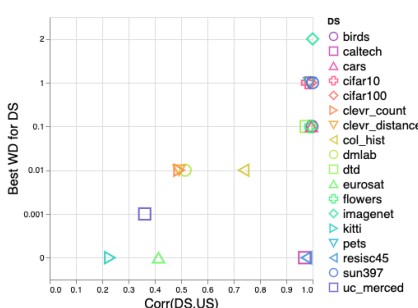

Figure D.4: Optimal weight decay as a function of rank correlation between the performance of US and DS for different DS tasks.

there is not much information in the head that will help in the DS performance and one can even remove the head and some of the top layers as seen in the analysis of Figure 6.

# E    ON THE GENERALIZATION OF OBSERVED PHENOMENA

The phenomena we described in the paper is not limited to the setting reported above. In this section, we discuss that the observations are robust to several changes in the setting.

**Number of shots:**    The DS-vs-US performance saturation phenomena and effect of head hyperparameters (WD, LR) are robust to the number of shots in the downstream task. This can be seen in Figures D.3, F.25, and  F.26.

**Transfer vs. few-shot:**    In addition to robustness to number of shots in few-shot setting, the reported phenomena is consistent across both few-shot and fine-tuning setting (aka transfer learning). Note that this is not direct implication of the previous assertion. In few-shot setting we keep the network weights fixed and only replace the head and train it for the downstream task. In fine-tuning

setting however, weights from the rest of the network are also updated using the training data for the downstream task. Figure E.1 presents the results of the effect of scaling in the fine-tuning setup on VTAB Benchmark. Note that VTAB considers a low-sample regime (1000-examples), which reflects performance under a reasonable labelling budget. Results in Figure E.1 correspond to the same controlled experiments that we performed in from Figure 4 for few-shot setting. Results on VTAB as well as VTAB subsets, i.e., natural, specialized, and structured shows similar general trends to few-shot setup. See Appendix H for additional observations and more detailed results in the fine-tuning setup.

**Scaling of plots:** Many of the works that consider transfer performance accuracy or how model accuracy changes by scaling up (Kornblith et al., 2019; Kaplan et al., 2020; Hernandez et al., 2021), scale the accuracy by passing it through a logit transformation ($\text{logit}(p) = \log(\frac{p}{1-p}) = \log(p) - \log(1-p)$, i.e., instead of plotting accuracy, they plot logit of accuracy. Logit function (which is the inverse of sigmoid function) has the drawback of being sensitive to low values. Meaning that if we plot a range of values that includes values close to zero, the logit plot is mainly influenced by values between 0 and 0.15 and the bigger values are collapsed mostly on top of each other. To mitigate this sensitivity, one can instead plot the the second term $-\log(1-p)$. We considered both these scaling options as well as not scaling the accuracies and observed that both phenomena presented in the paper are robust to choice of scaling. For corresponding plots to logit and $-\log(1-p)$ see Figure F.2 in Appendix F.1.

**Architecture:** In this work, we investigated a family of architectures and a number of different architectural changes in different models from the Vision Transformers, MLP-Mixers and ResNets. It has been widely inspected that in large data regime the role of inductive biases and architecture-specific parameters diminishes. Moreover, there is evidence indicating that choice of architecture does not impact the power law governing the DS performance (Kaplan et al., 2020; Taori et al., 2020; Miller et al., 2021). This is also observed in (Kornblith et al., 2019) that effect of architecture is only observed through US performance. Therefore, we expect that our results generalizes to other large scale architectures such as ResNet-151 and EfficientNet (Tan & Le, 2019) (that is made of CNN blocks).

# F  ADDITIONAL FIGURES

## F.1  ADDITIONAL FIGURES FOR SECTION 2.2

Figure F.1 depicts a similar plot to Figure 1 with all the 4800 experiments (pre-trained on JFT or ImageNet21K), for 1 and 25 shots.

Figure F.2 presents a scaled version of Figure 1, given the scaling of downstream accuracies, discussed in Section E.

### F.1.1  DETAILS THE EXPERIMENTAL SETUP FOR FITTING EQUATION 1

Figures F.6 and F.7 illustrate the fitted curves to the the convex hull and all data points in the US-vs-DS accuracy plots respectively. We use the points from the lower US accuracies (0.0, 0.45) as fitting data and higher US accuracies (0.45-0.50) as held out data to fit equation 1. For the convex hull fit, we first compute the convex hull of the given data points, and find the fit to the convex hull. In Figure F.8 and F.9, we compare the fitted curves when we fit equation 1 to all data points or the convex hull of all data points for 1 shot and 25 shot.

To measure the sensitivity of the predictive power of the fitted equation to the number of samples, we conduct the experiment with different number of data points sampled randomly (uniform distribution across all data points), and for each sample size we repeat the experiment 10 times (where we take a new sample for each trial). We use the points from higher US accuracies as held out data. Prediction error captures the difference between power law prediction and observed value of the DS accuracy. Fitting error captures the difference of power law values from the points that are used in calculating power law parameters. We plot fitting error and prediction error as the number of samples changes. Figures F.11, F.12, F.13, F.14, F.15 and F.16 depict the mean prediction error and mean fitting error for each sample size as well as their standard deviation across the 10 trial.

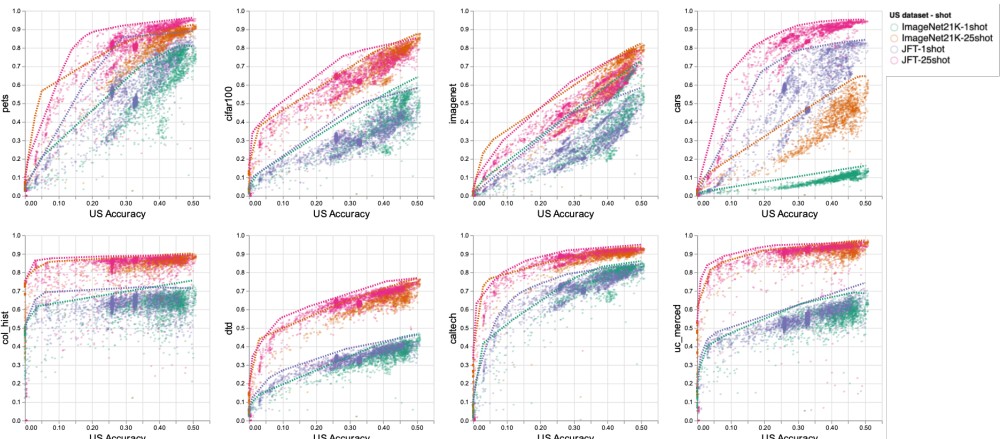

Figure F.1: Performance of 8 different downstream tasks vs upstream based on more than 4800 different experiments (2974 Vision Transformers, 1593 MLP-Mixers , 249 ResNets). The experiments are grouped based on the upstream dataset (JFT or ImageNet21K) and number of shots in few-shot evaluation (1 and 25). The dotted line shows the convex hull of points on the DS-vs-US plot. The fact that saturation happens at different values for the two upstream tasks suggests that saturation does not solely depend on DS task, rather, it is about the relationship between US, DS tasks.

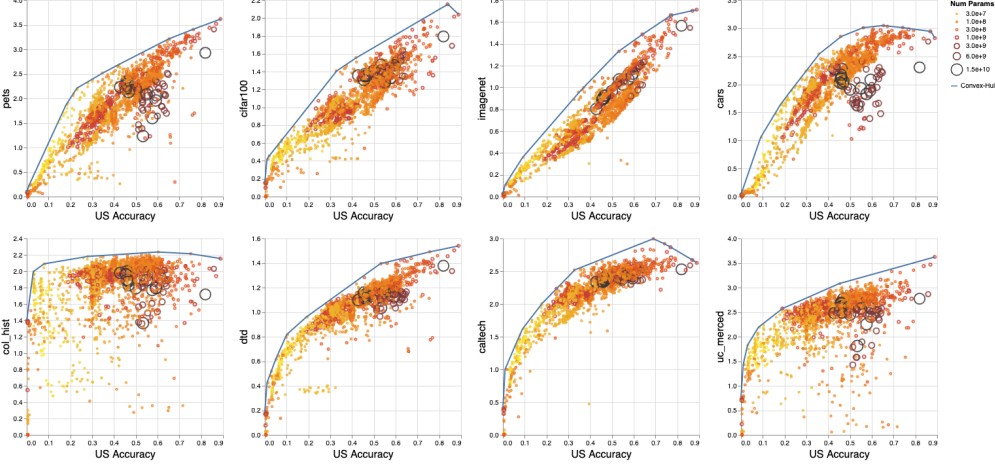

Figure F.2: Performance of upstream vs downstream (8 different tasks) based on more than 3K different ViT models with different configurations, pre-trained on JFT and evaluated on few-shot (25 shots), where downstream accuracies are scaled using $-\log(1-p)$.

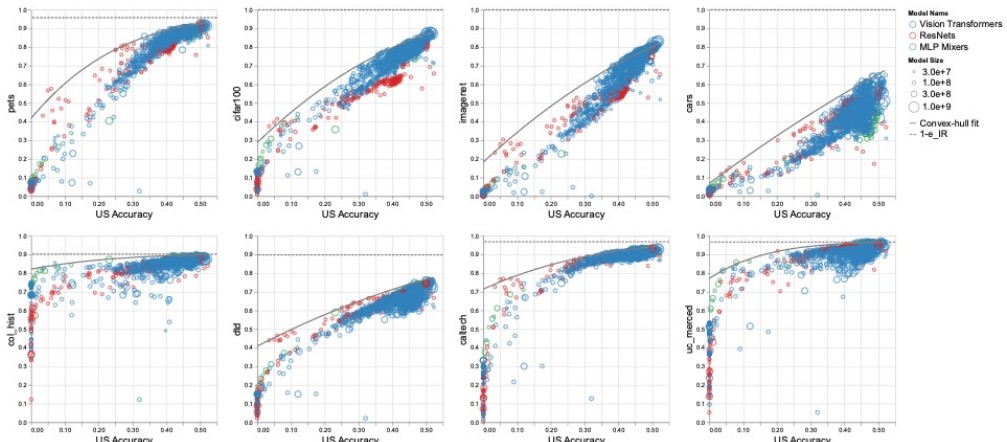

Figure F.3: The performance of downstream (8 different tasks) vs of upstream based on more than 1.4k different Vision Transformers, 90 MLP mixers and 233 ResNets, with different configurations. The models are pre-trained on ImageNet21K and evaluated in few-shot settings (25 shots). *As the upstream performance improves, the downstream performance saturates.* Even if US accuracy reaches 100% accuracy, the DS accuracy may not reach the 100% accuracy and saturates at a lower value. We observe a non-linear relationship between upstream and downstream accuracy and model the relationship with a power law function to predict the downstream performance given the upstream performance. The plot also show horizontal line which is the predicted downstream accuracy if upstream accuracy reaches 100%.

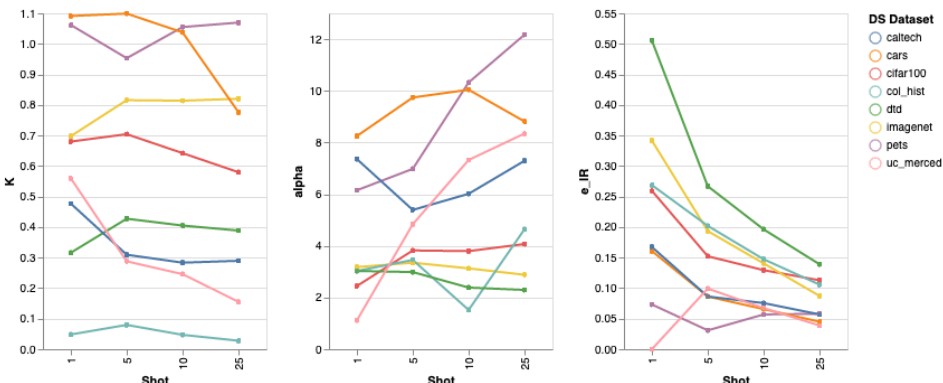

Figure F.4: Effect of number shots and DS task on the value of parameters of the power law curves, upstream task is JFT. We note that DS task affect all parameters, while number of shots mostly impacts $k$ and $e_{\text{IR}}$.

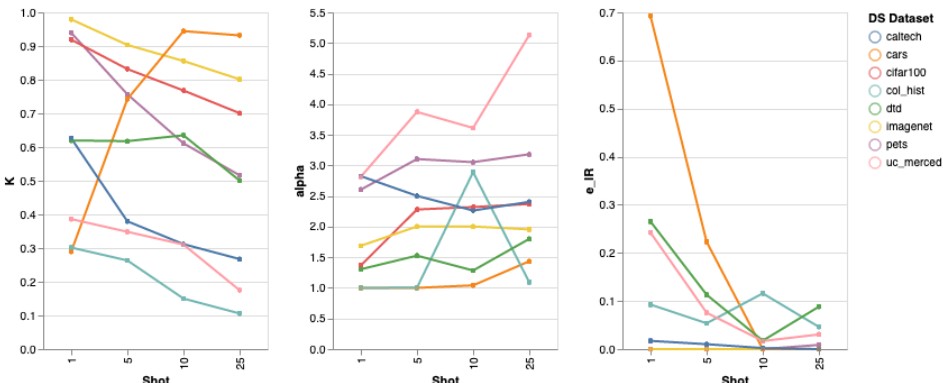

Figure F.5: Effect of number shots and DS task on the value of parameters of the power law curves, upstream task is ImageNet 21k. We note that DS task affect all parameters, while number of shots mostly impacts $k$ and $e_{\text{IR}}$.

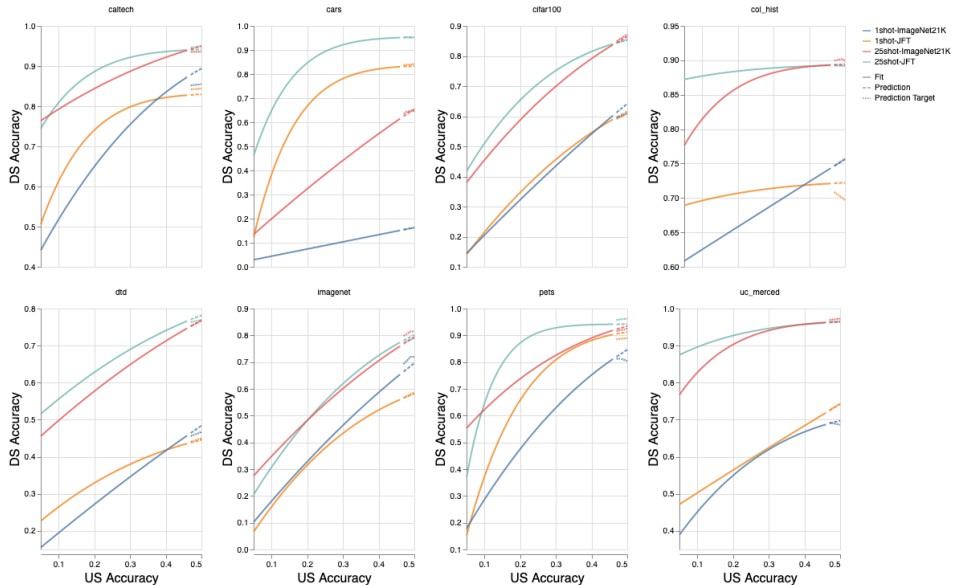

Figure F.6: Power law curves that are fitted to the points on the convex hull corresponding to experiment results from Figure F.1. We plot the predictions from the power law curve on the higher US accuracies to the ground truth (prediction target) and observe that power law curve closely predicts the performance of DS. .

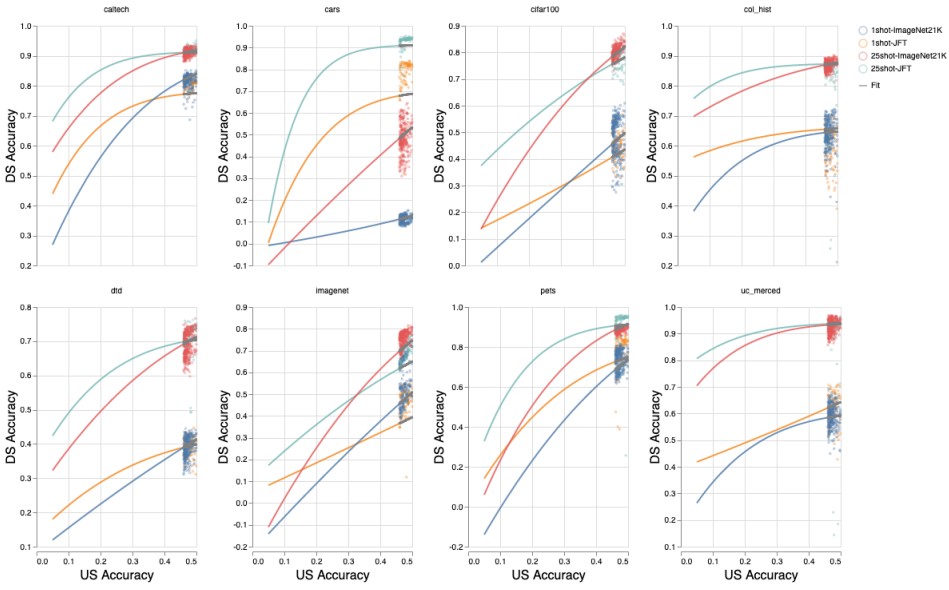

Figure F.7: Power law curves that are fitted to all point corresponding to experiment results from Figure F.1. We plot the predictions from the power law curve on the higher US accuracies to the ground truth (prediction target) and observe that power law curve closely predicts the performance of DS.

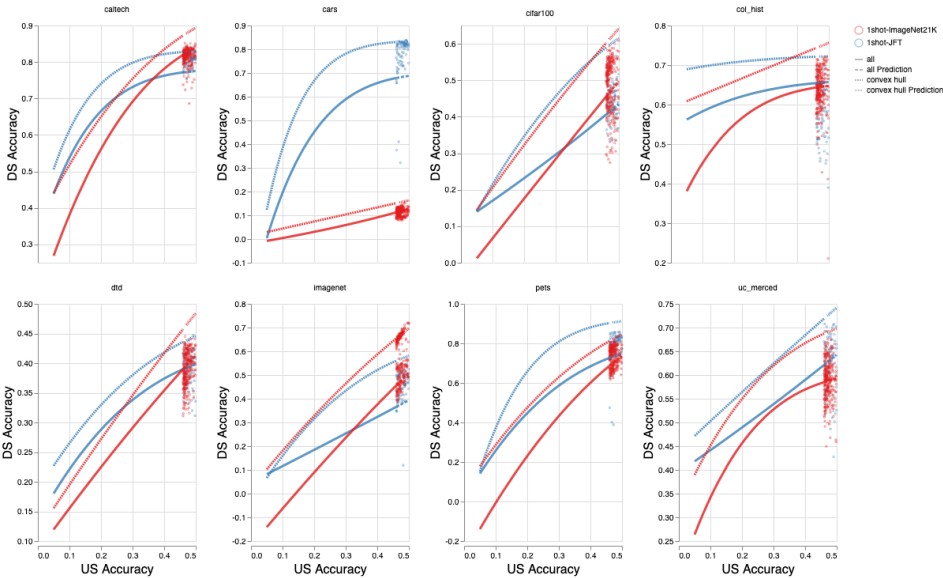

Figure F.8: Comparing fitted curves when we use convex hull (Figure F.6) vs when we use all samples (Figure F.7), when number of shots is 1.

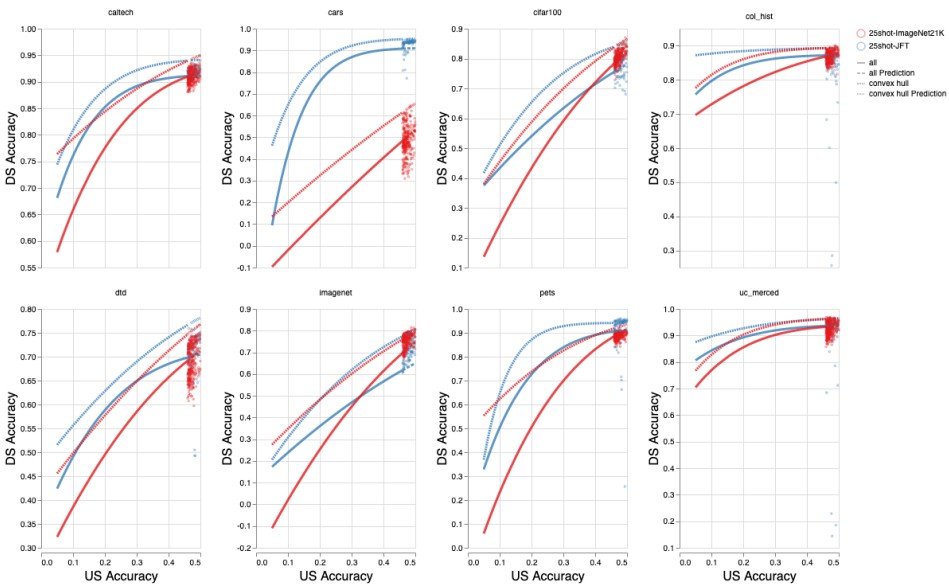

Figure F.9: Comparing fitted curves when we use convex hull (Figure F.6) vs when we use all samples (Figure F.7), when number of shots is 25.

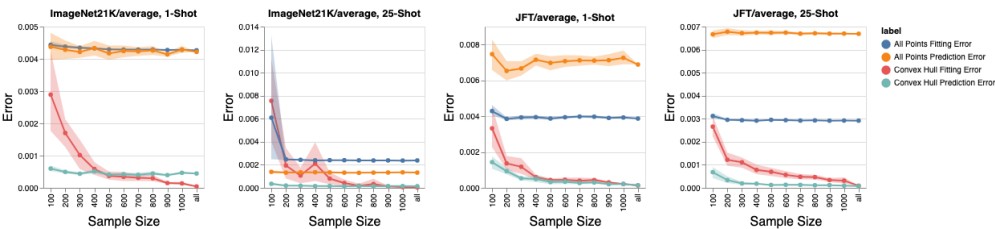

Figure F.10: The effect of sample size on power law curves. The curves are fitted to the convex hull of experiments as well as all data points from Figure F.1. We use the points from higher US accuracies as held out data. Prediction error captures the difference between power law prediction and observed value of the DS accuracy. Fitting error captures the difference of power law values from the points that are used in calculating power law parameters. We plot fitting error and prediction error as the number of samples changes. For more details see Appendix F.1.1.

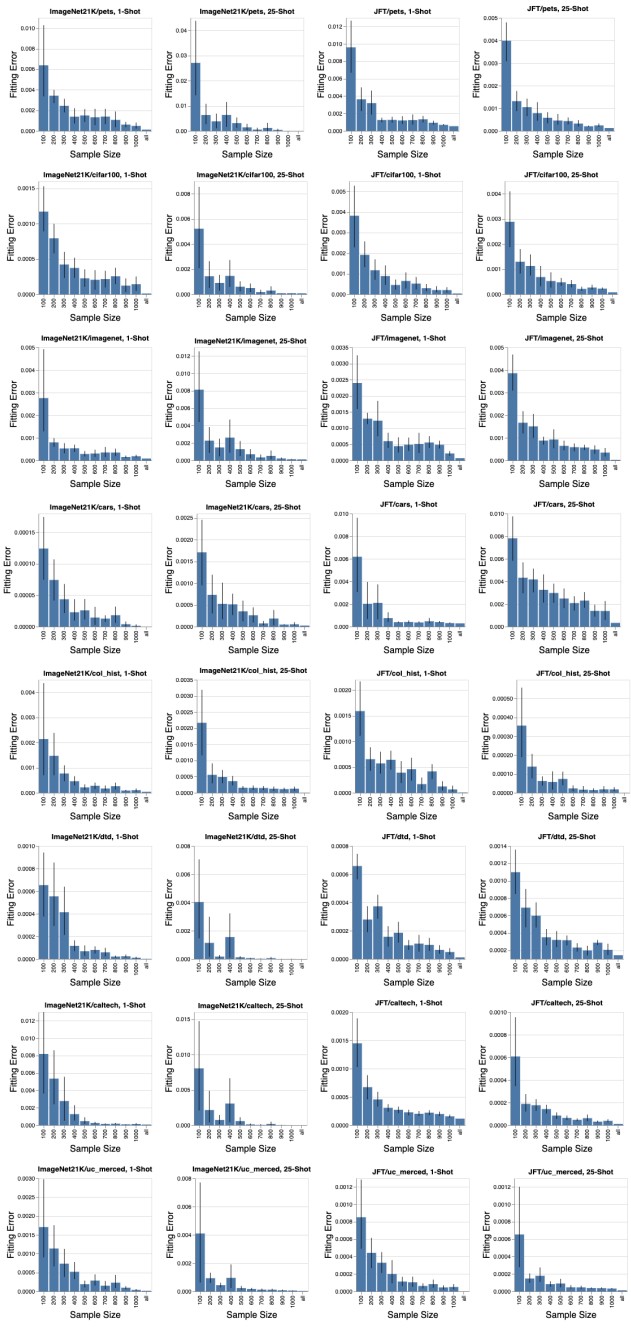

Figure F.11: Effect of sample size when fitting the power law to convex hull of the samples on the average fitting error.

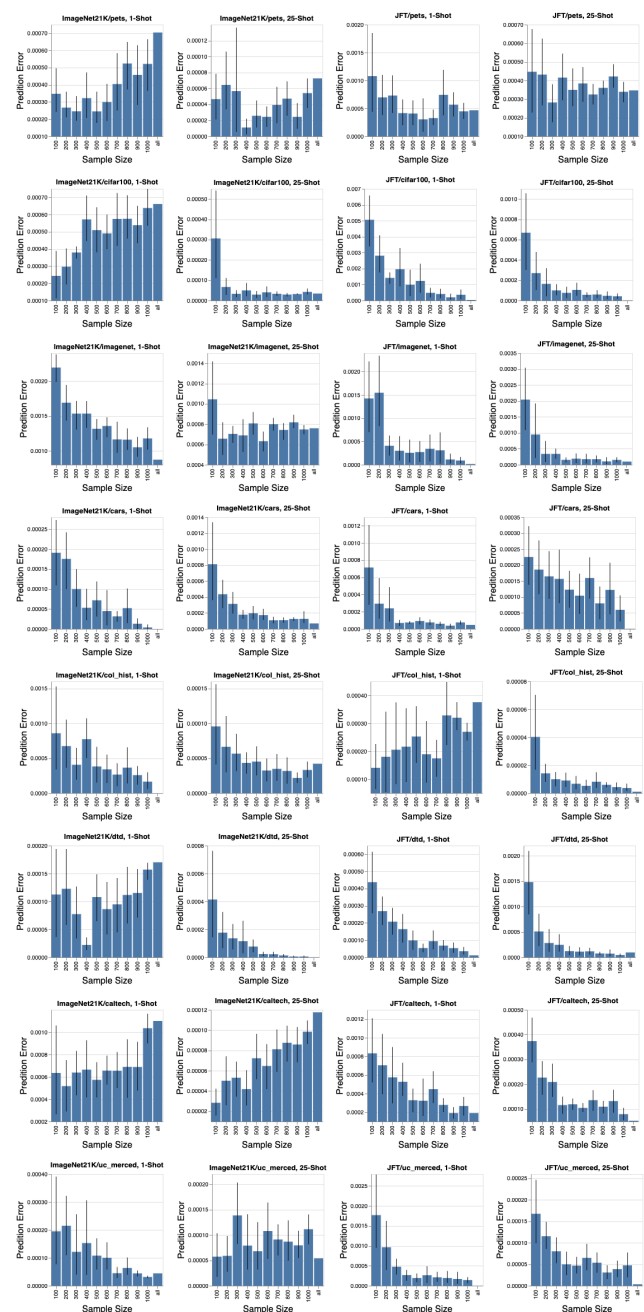

Figure F.12: Effect of sample size when fitting the power law to convex hull of the samples on the average prediction error.

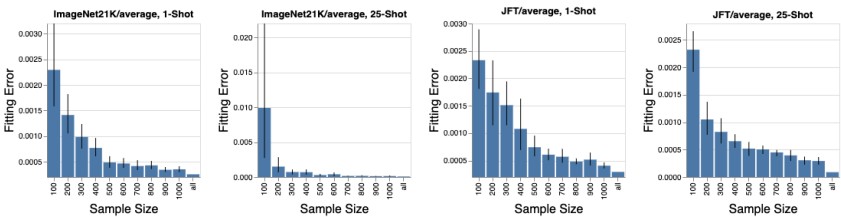

Figure F.13: Effect of sample size when fitting the power law to convex hull of the samples on the average fitting error.

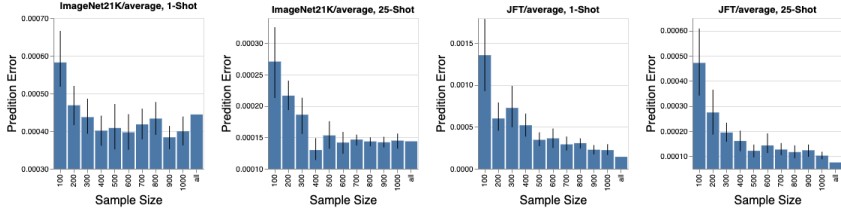

Figure F.14: Effect of sample size when fitting the power law to convex hull of the samples on the average prediction error.

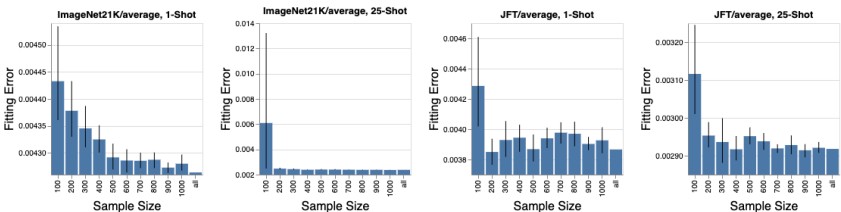

Figure F.15: Effect of sample size when fitting the power law to all samples on the average fitting error.

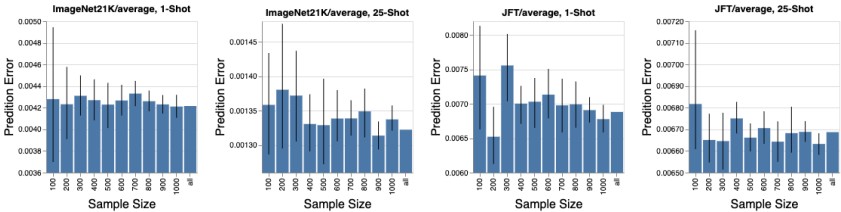

Figure F.16: Effect of sample size when fitting the power law to all samples on the average prediction error.

| DS | US | Parameter | Correlation with Number of Shots |
|---|---|---|---|
| caltech | ImageNet21K | K | -0.777892 |
| caltech | ImageNet21K | $\alpha$ | -0.582066 |
| caltech | ImageNet21K | $e_{\text{IR}}$ | -0.845368 |
| caltech | JFT | K | -0.620526 |
| caltech | JFT | $\alpha$ | 0.259305 |
| caltech | JFT | $e_{\text{IR}}$ | -0.762856 |
| cars | ImageNet21K | K | 0.720391 |
| cars | ImageNet21K | $\alpha$ | 0.960490 |
| cars | ImageNet21K | $e_{\text{IR}}$ | -0.737273 |
| cars | JFT | K | -0.976599 |
| cars | JFT | $\alpha$ | -0.034033 |
| cars | JFT | $e_{\text{IR}}$ | -0.809016 |
| cifar100 | ImageNet21K | K | -0.918914 |
| cifar100 | ImageNet21K | $\alpha$ | 0.683485 |
| cifar100 | ImageNet21K | $e_{\text{IR}}$ | -0.587304 |
| cifar100 | JFT | K | -0.934455 |
| cifar100 | JFT | $\alpha$ | 0.707966 |
| cifar100 | JFT | $e_{\text{IR}}$ | -0.754030 |
| col_hist | ImageNet21K | K | -0.756297 |
| col_hist | ImageNet21K | $\alpha$ | 0.947101 |
| col_hist | ImageNet21K | $e_{\text{IR}}$ | -0.104776 |
| col_hist | JFT | K | -0.534724 |
| col_hist | JFT | $\alpha$ | 0.466138 |
| col_hist | JFT | $e_{\text{IR}}$ | -0.848960 |
| dtd | ImageNet21K | K | -0.892400 |
| dtd | ImageNet21K | $\alpha$ | 0.810935 |
| dtd | ImageNet21K | $e_{\text{IR}}$ | -0.532797 |
| dtd | JFT | K | 0.392218 |
| dtd | JFT | $\alpha$ | -0.751290 |
| dtd | JFT | $e_{\text{IR}}$ | -0.806674 |
| imagenet | ImageNet21K | K | -0.923350 |
| imagenet | ImageNet21K | $\alpha$ | 0.464193 |
| imagenet | ImageNet21K | $e_{\text{IR}}$ | -0.590325 |
| imagenet | JFT | K | 0.618935 |
| imagenet | JFT | $\alpha$ | -0.866692 |
| imagenet | JFT | $e_{\text{IR}}$ | -0.847294 |
| pets | ImageNet21K | K | -0.895292 |
| pets | ImageNet21K | $\alpha$ | 0.707198 |
| pets | ImageNet21K | $e_{\text{IR}}$ | 0.936508 |
| pets | JFT | K | 0.398171 |
| pets | JFT | $\alpha$ | 0.937076 |
| pets | JFT | $e_{\text{IR}}$ | -0.003738 |
| uc_merced | ImageNet21K | K | -0.986538 |
| uc_merced | ImageNet21K | $\alpha$ | 0.942120 |
| uc_merced | ImageNet21K | $e_{\text{IR}}$ | -0.724245 |
| uc_merced | JFT | K | -0.821492 |
| uc_merced | JFT | $\alpha$ | 0.743757 |
| uc_merced | JFT | $e_{\text{IR}}$ | 0.019906 |

Table F.1: Correlation of each parameter with number of shots

| Parameter | Correlation for US=JFT | Correlation for US= ImageNet 21k |
|:---:|:---:|:---:|
| $k$ | -0.65 | -0.81 |
| $\alpha$ | 0.60 | 0.75 |
| $e_{\text{IR}}$ | -0.88 | -0.79 |

Table F.2: The Likelihood that each of the parameters of the scaling law increases/decreases as the number of shots increases, average over all DS tasks.

## F.2 ADDITIONAL FIGURES FOR SECTION 3

Figure F.17 shows the effect of scaling model, data, and compute on all downstream tasks. This is a more complete version of Figure 4 in the main paper that includes all 25 different downstream tasks.

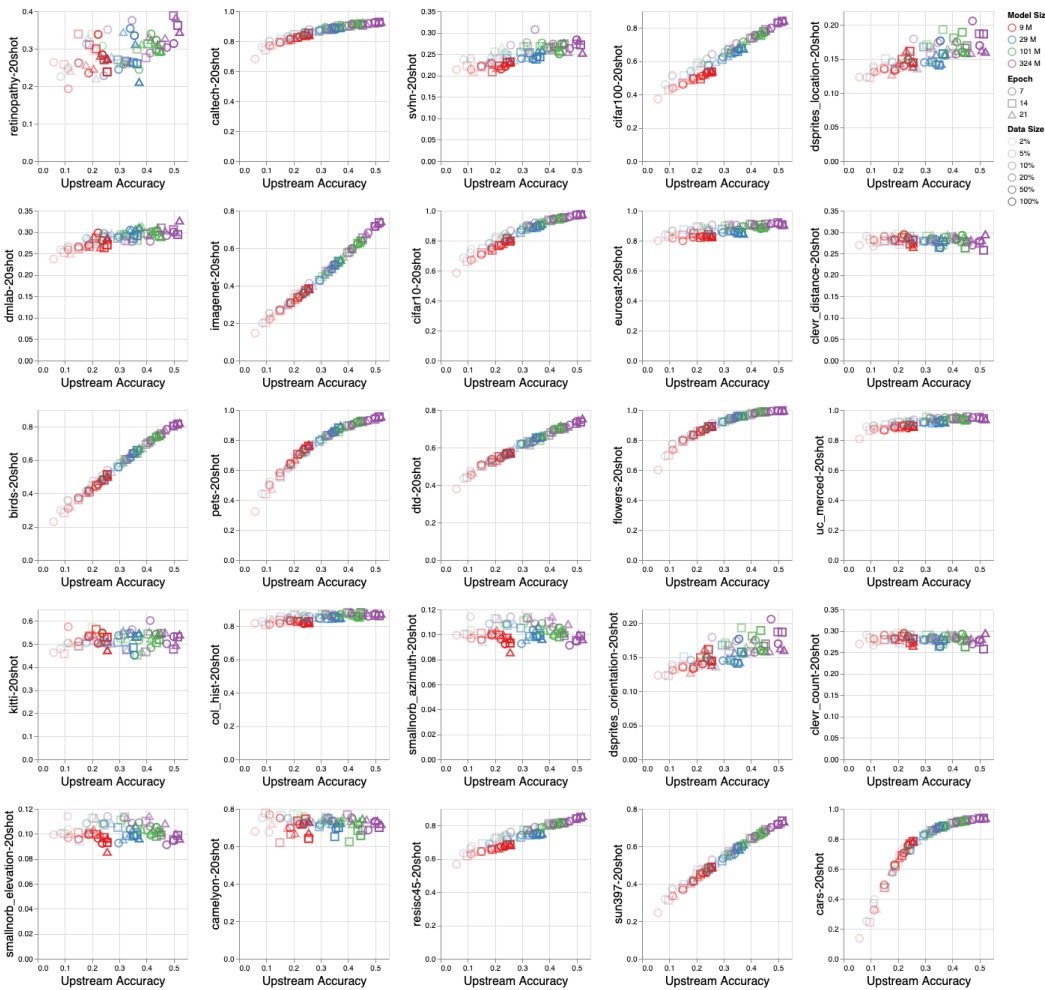

Figure F.17: Effect of controlled scale up with respect to the model size (number of parameters), data size (portion of the pretrained data), and compute (epochs) on 25 different downstream tasks in few-shot setup (20-shots).

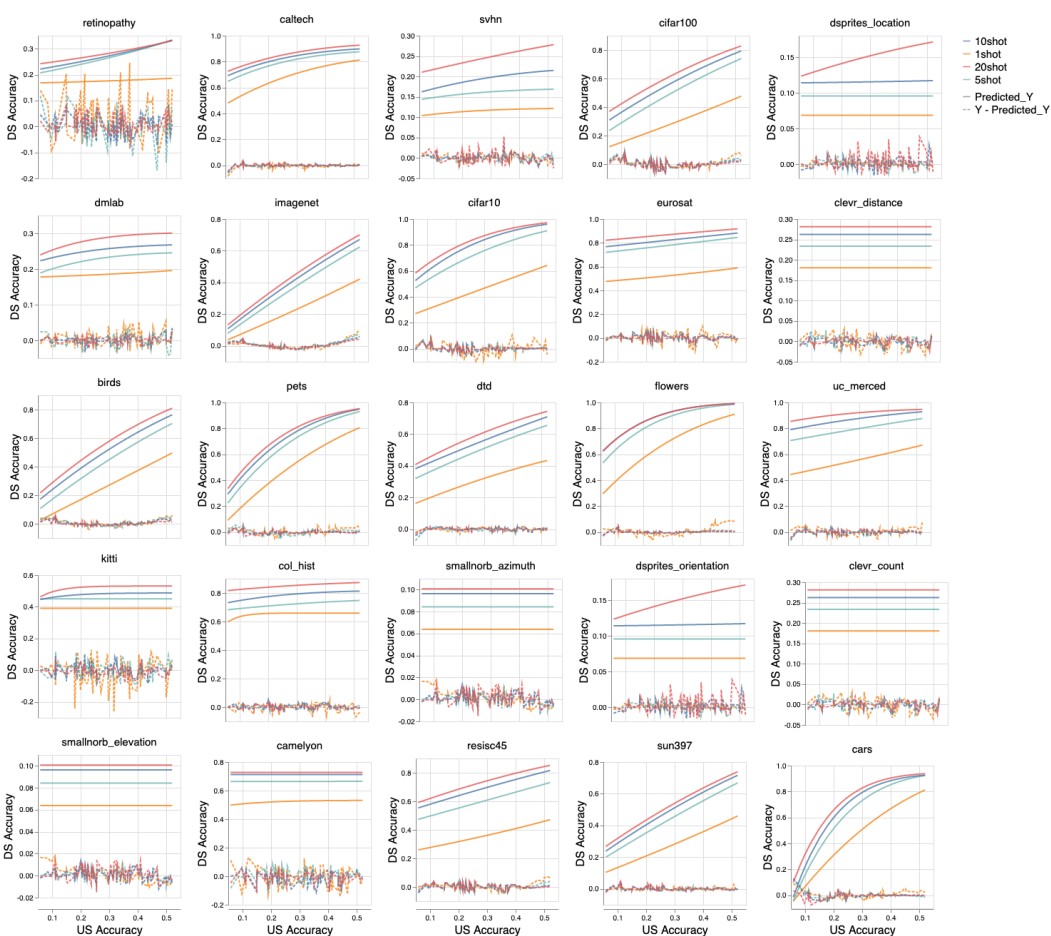

Figure F.18: Fitting the scaling law to points plotted Figure F.17 and depicting the value of error incurred in predicting DS accuracy.

| DS | $\sqrt{\sum (Y - Y')^2}$ |
|---|---|
| birds | 0.154270 |
| caltech | 0.102052 |
| camelyon | 0.402138 |
| cars | 0.197948 |
| cifar10 | 0.235078 |
| cifar100 | 0.242331 |
| clevr_count | 0.093481 |
| clevr_distance | 0.093481 |
| col_hist | 0.155221 |
| dmlab | 0.126028 |
| dsprites_location | 0.059326 |
| dsprites_orientation | 0.059326 |
| dtd | 0.088551 |
| eurosat | 0.258027 |
| flowers | 0.141492 |
| imagenet | 0.188222 |
| kitti | 0.438465 |
| pets | 0.141252 |
| resisc45 | 0.188155 |
| retinopathy | 0.446441 |
| smallnorb_azimuth | 0.049473 |
| smallnorb_elevation | 0.049473 |
| sun397 | 0.085017 |
| svhn | 0.082023 |
| uc_merced | 0.158118 |

Table F.3: Root squared error of predicted DS accuracy when fitting the points in Figure F.2 with Equation 1.

## F.3  ADDITIONAL FIGURES FOR SECTION 4

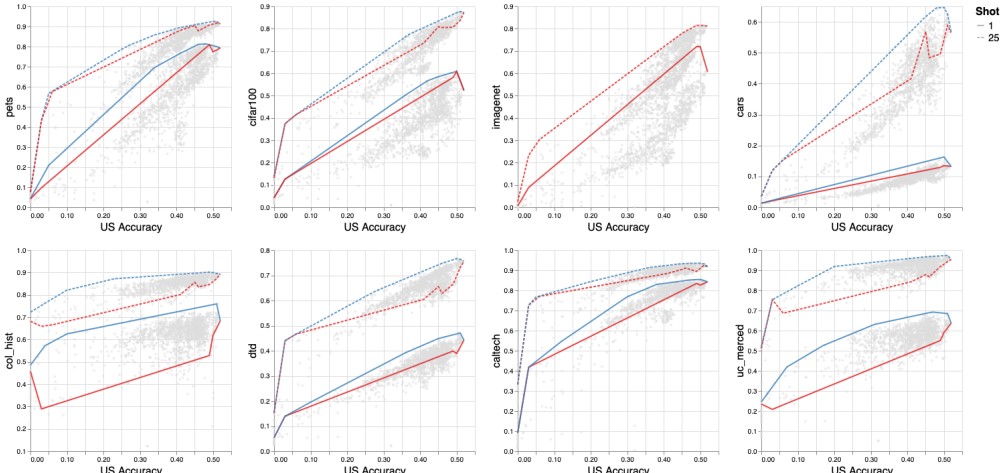

Figure F.19: The overlay the convex hull of ImageNet DS-vs-US plot on DS-vs-US plots of all DS tasks from Figure 1. US task is ImageNet21K. We observe that Best performing ImageNet models perform very similarly to best performing models in several DS tasks but not all DS tasks. Moreover, as the US performance increases, the gap between best performing ImageNet models and best performing DS task models reduces significantly.

Figure F.20 depicts Spearman correlation between accuracies on different downstream tasks. Figure F.21 shows Spearman correlation between accuracies on different downstream tasks and the upstream task.

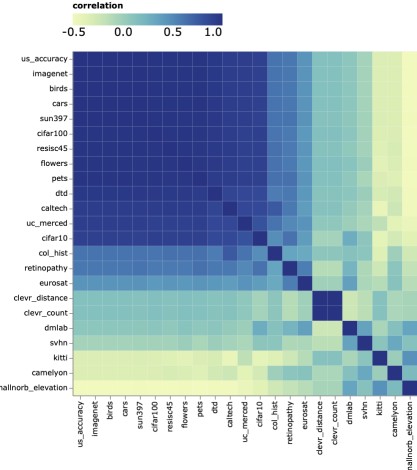

Figure F.20: Spearman correlation between accuracies on different downstream tasks.

Figure F.22 illustrates the quality of representations from different layers on all downstream tasks. This is a more complete version of Figure 6 in Section 4 that includes all 25 different downstream tasks.

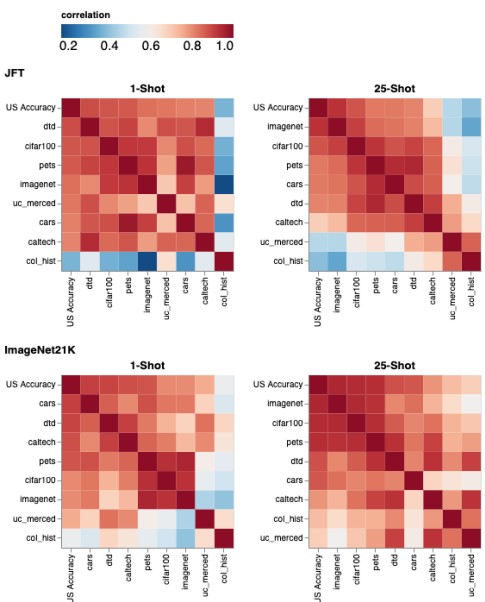

Figure F.21: Spearman correlation between accuracies on different downstream tasks and the upstream task, based on more than 3K different ViT models with different configurations.

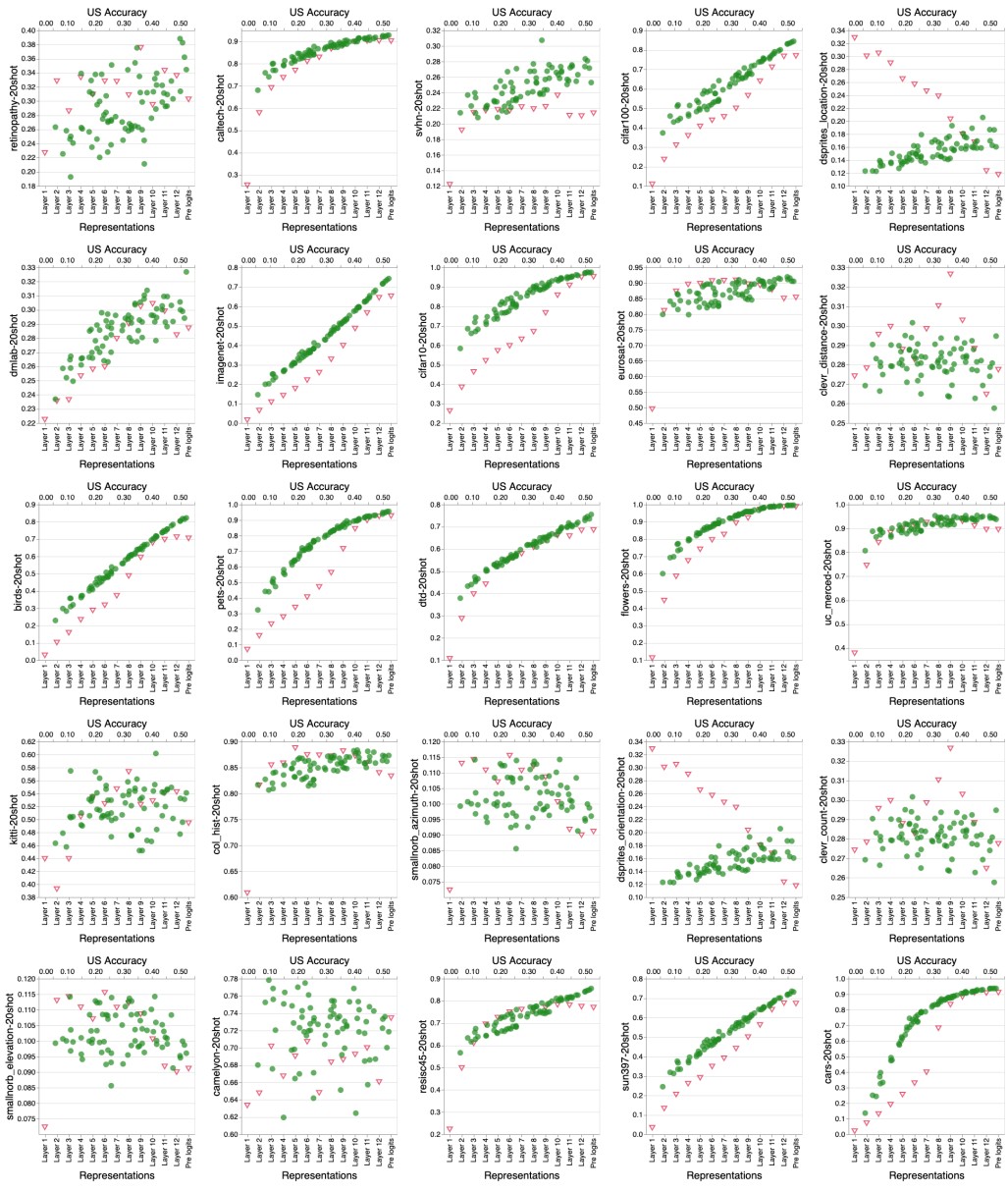

Figure F.22: Investigating the effect of choosing representations from different layers on the downstream tasks performance overlay-ed with with the effect of scaling (model, data, and compute) on downstream performance when upstream task is JFT. The red triangles in the plots are performance on downstream task when representation used in the few-shot learning is from different layers of the model. The green circles in the plots overlay the US versus DS performance of different experiments from Figure F.17 on each task. Here we sketch the plots for 25 downstream tasks.

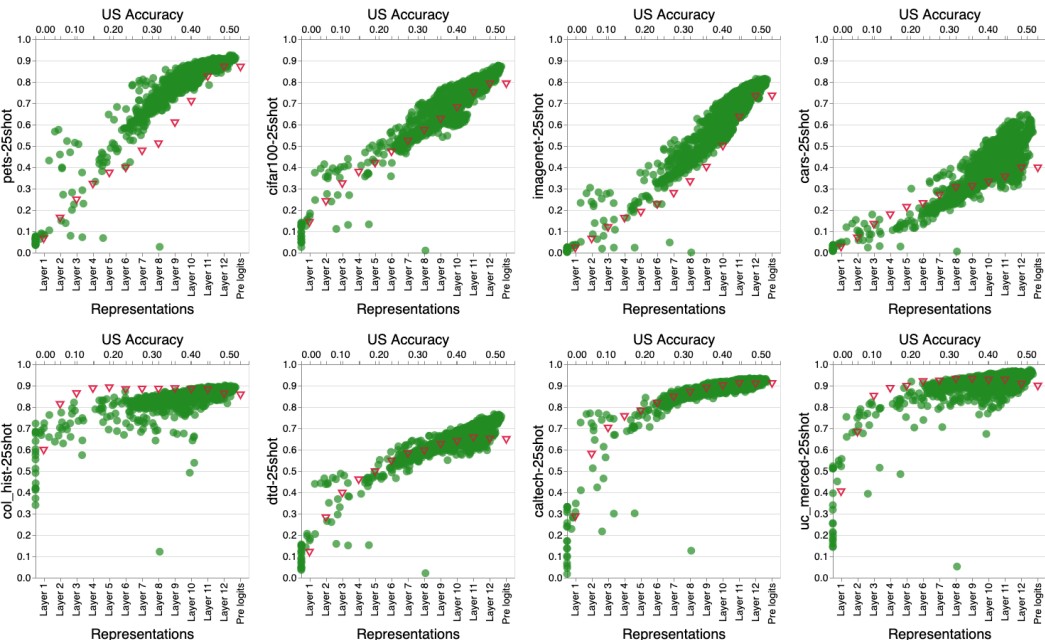

Figure F.23: Investigating the effect of choosing representations from different layers on the downstream tasks performance overlay-ed with US-vs-DS performance when upstream task is ImageNet21K. The red triangles in the plots are performance on downstream task when representation used in the few-shot learning is from different layers of the model. The green circles in the plots overlay the DS versus US performance of different experiments from Figure F.3 on each task. Red triangles use the x-axis on the bottom and the green circles use the x-axis on the top. We note that for those DS tasks that are similar to US, such as ImageNet, the higher the representation layer the better performance on DS. On the contrary, For those DS tasks that saturate fast, such as UC-Merced and col_hist, the optimal layer is not the last one and the model can be cut at lower layers leading to better performance.

### F.4 ADDITIONAL FIGURES FOR APPENDIX C

Figure F.24 illustrates the effect of increasing head weight decay on all downstream tasks. Figure D.1 in the main paper that includes all downstream tasks.

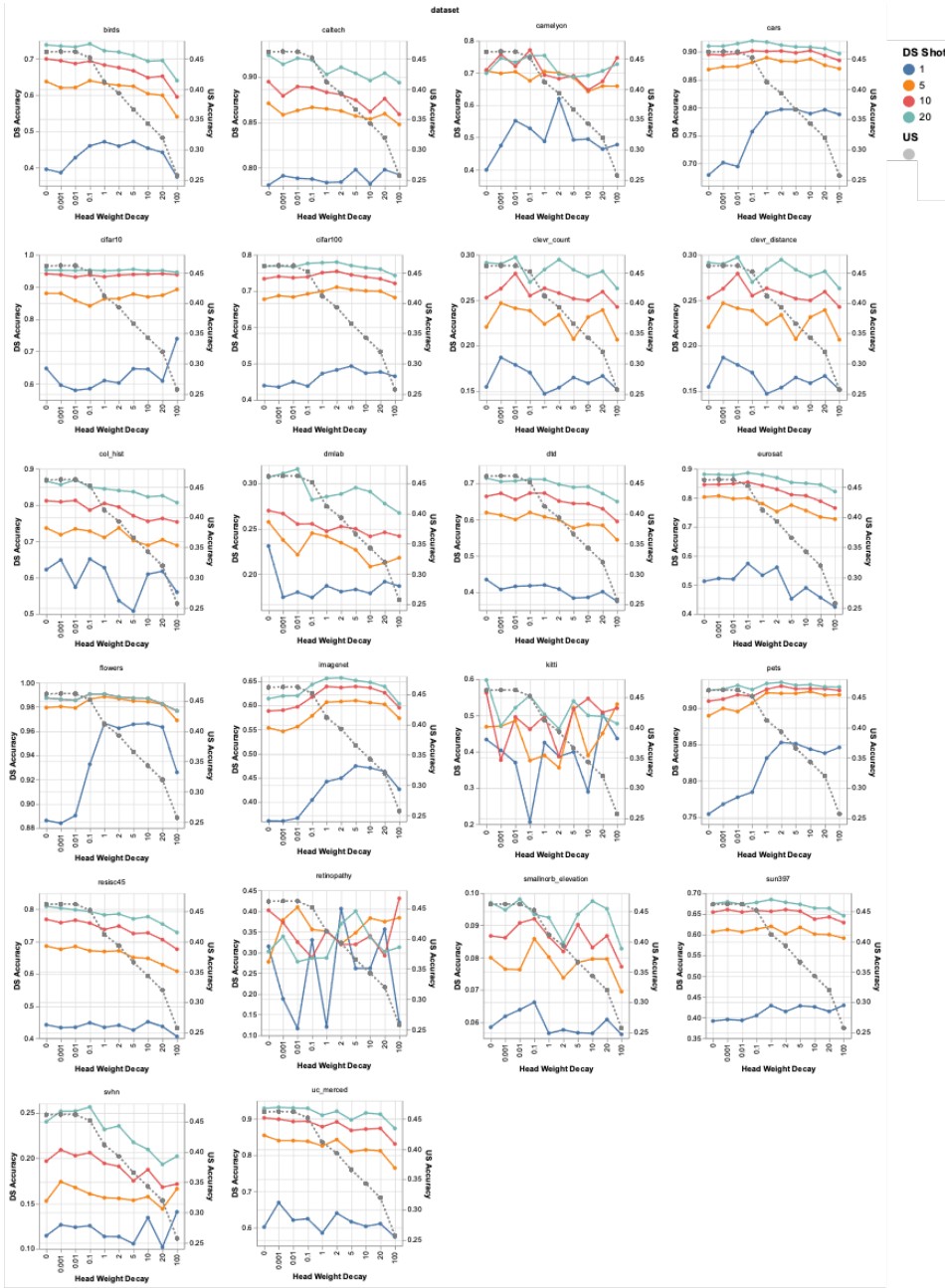

Figure F.24: The effect of increasing head weight decay in the performance of upstream verses downstream (all shots, all datasets).

Figure F.25 show the best head weight decay for all downstream tasks. This figure is a more complete version of Figure D.3.

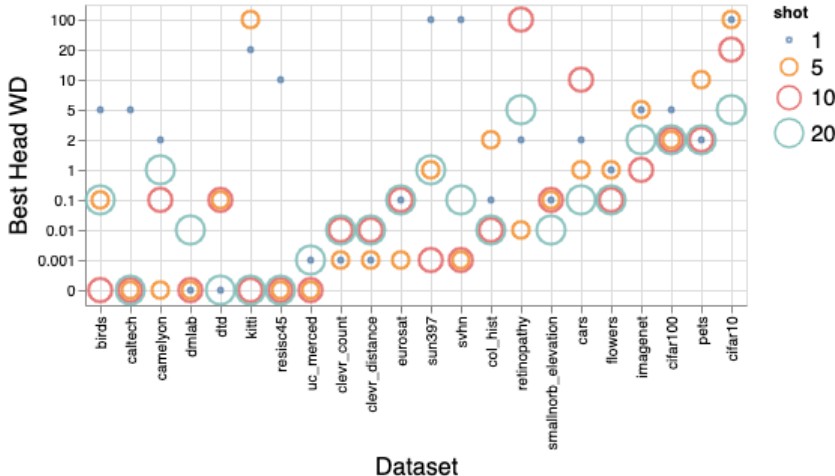

Figure F.25: Optimum head weight decay for different downstream tasks and for different number of shots.

Figure F.26 illustrates the effect of changing head weight decay on all downstream tasks, when we train longer (for 14 epochs instead of 7 that is reported in Figure D.1). The changes is consistent across different epochs as well as different number of shots.

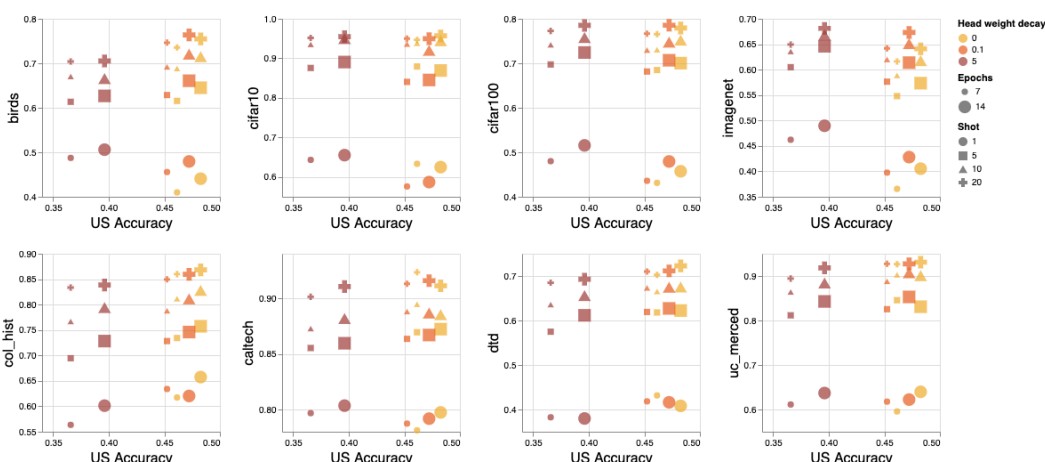

Figure F.26: The effect of changing head weight decay when trained for 7 or 14 epochs for different number of shots.

Figure F.30 illustrates the effect of increasing head weight decay on pre-logit layer margin for all downstream tasks.

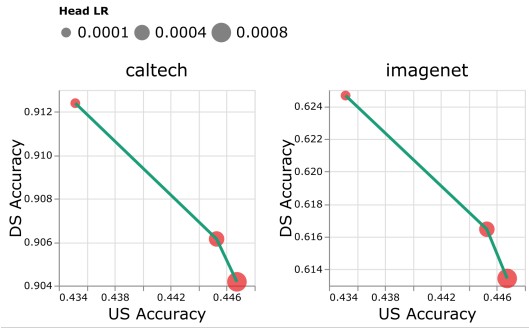

Figure F.27: The effect of increasing head learning rate in performance of upstream (JFT) versus performance of downstream (ImageNet1k and Caltech101)

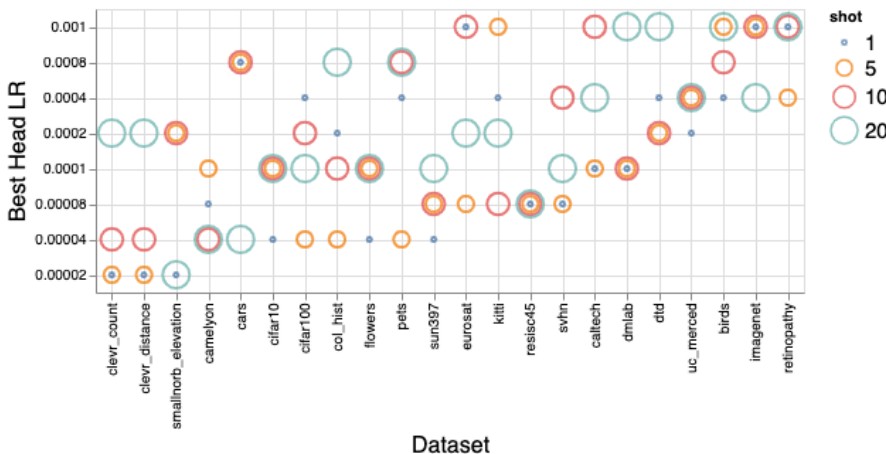

Figure F.28: Optimum head learning rate for different downstream tasks and for different number of shots.

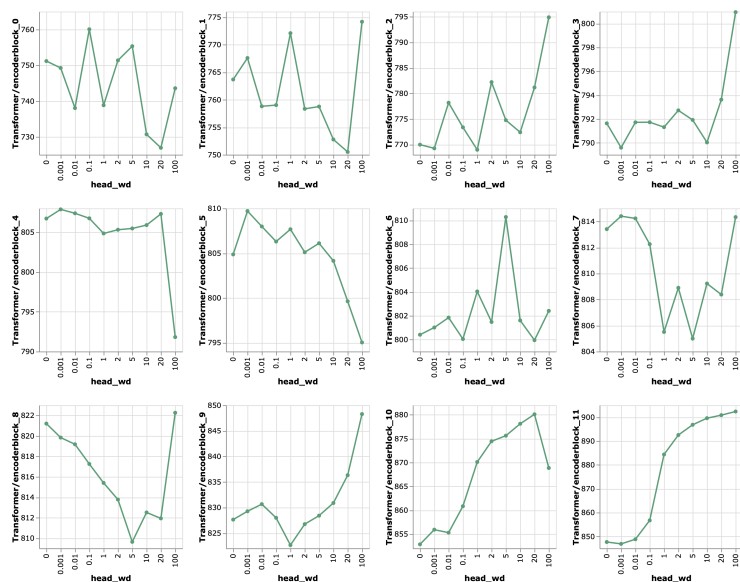

Figure F.29: L2 Norm of different layers of the ViT model for different values of head weight decay.

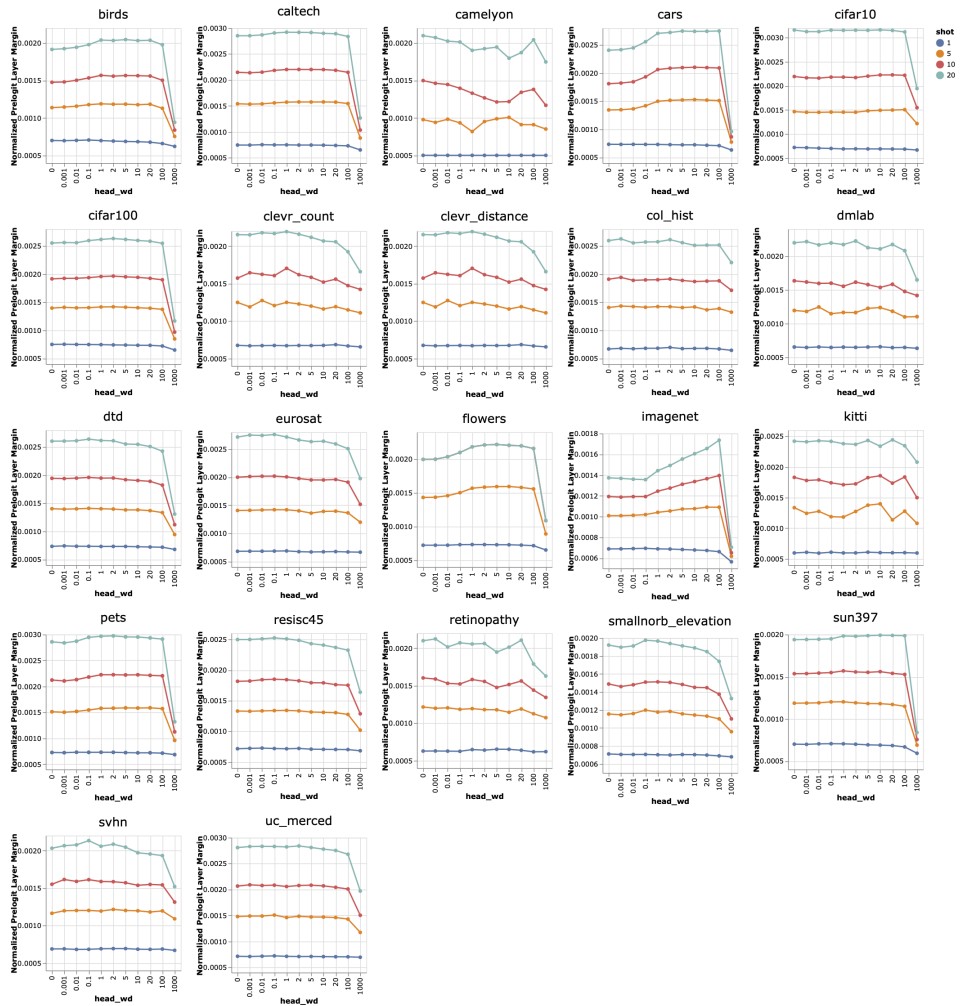

Figure F.30: The effect of increasing head weight decay in the pre-logit layer margin for downstream (all shots, all datasets). In this plot L2 term for downstream few-shot classifiers is set to 4096.

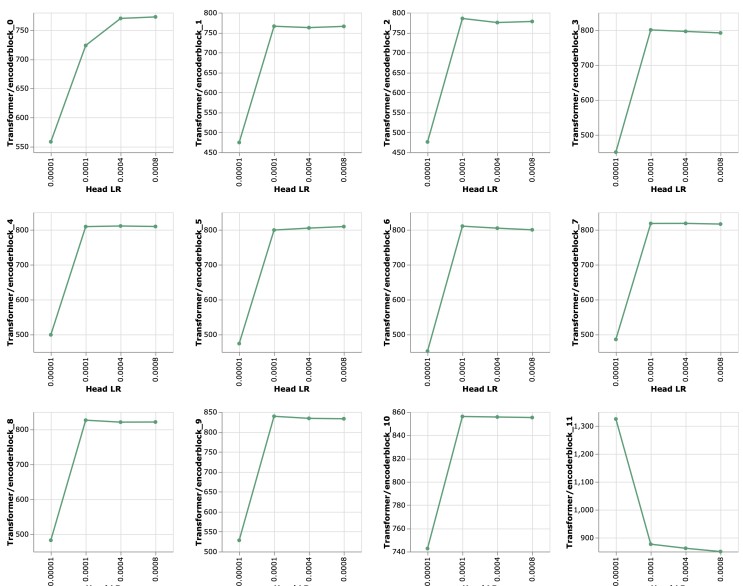

Figure F.31: L2 Norm of different layers of the ViT model for different values of head learning rate.

# G  EXPERIMENT SETUP

## G.1  CONFIGURATIONS OF THE MODELS USED IN THE LARGE SCALE META-STUDY

We investigate more than 4800 experiments with Vision Transformers, MLP-mixers and ResNets with different configurations (2974 Vision Transformers, 1593 MLP-Mixers , 249 ResNets.), when pre-trained on a large amount of data in a supervised fashion and evaluated on several downstream image recognition tasks through few-shot learning and fine tuning. We note that while there are much fewer ResNet experiments in the meta data we have collected, these are the best performing ResNet models as researchers know how to tune hyper-parameters for these class of models to achieve best performance. However, for Vision Transformers and MLP-Mixers, best practices for hyper-parameter tuning is yet to be figured out as these are newer architectures. In addition, our data suggests that the convex hull would not be affected significantly by having more ResNet models. Therefore, due to computational and environmental costs, we refrain from training many ResNets. These experiments vary in terms of the upstream dataset (either JFT-300M (Sun et al., 2017) with 303M images and 18k classes or ImageNet21K (Deng et al., 2009) with 14M images and 21k classes), model size and shape (different hyper-parameters of the architecture), optimization (e.g. different learning rate values and learning rate schedules, different weight decays, different optimizers), compute (e.g. number of epochs) and other knobs that researchers changed during development of models for various purposes.

In the experiments we run ourselves (Section 3), we mainly use ViT-B/32, which is the base model with patch size $32 \times 32$. We also have tiny (9.4687e+6 parameters), small (2.9536e+7 parameters), base (1.0152e+8 parameters) and large (3.2426e+8 parameters) models for the controlled scaling experiments. We pre-train our models on JFT for 7 epochs and evaluate on more than 20 tasks. For the downstream evaluation, we mainly focus on few-shot learning setup (1, 5, 10, and 20 shots) as well as fine-tuning for some of the ablations. This is motivated by the fact that the effect of transfer learning vanishes as the number of downstream data points increases (Kornblith et al., 2019; Zoph et al., 2020; Mensink et al., 2021). Hence, we focus on a setting where transfer learning shines the most. In both aggregated and controlled experiments, in the few-shot setup, a linear classifier is trained on top of the representations from the frozen pre-trained model, given only a fixed number of training examples per class. In the fine-tuning setup, we follow VTAB standard (Zhai et al., 2019) and use 1000 training samples from the downstream task and update all the parameters of the model besides the downstream head. The details on upstream and downstream task benchmarks and training setup appear in the Appendix G.

## G.2  TRAINING DETAILS

For the controlled experiments, we train all models using Adam (Kingma & Ba, 2014) with $\beta_1 = 0.9$, $\beta_2 = 0.999$. In all experiments, the batch size is set to $4096$. The deffault weight decay used in the experiments is $0.1$, unless the changed value is mentioned in the description of the experiment. For the learning rate, we se the value to $8e - 4$ (unless for large models that we use $4e - 4$) and use a linear decay, with a warmup of $1000$ steps.

## G.3  DATASETS

Table G.4 summarizes the datasets used in our experiments.

Table G.4: Summary of datasets used in our experiments, part I

| Dataset | Description | Reference |
|---------|-------------|-----------|
| ImageNet | 1.28M labelled natural images. | (Deng et al., 2009) |
| Caltech101 | The task consists in classifying pictures of objects (101 classes plus a background clutter class), including animals, airplanes, chairs, or scissors. The image size varies, but it typically ranges from 200-300 pixels per edge. | `http://www.vision.caltech.edu/Image_Datasets/Caltech101/` |
| CIFAR-10 | The task consists in classifying natural images (10 classes, with 6000 training images each). Some examples include apples, bottles, dinosaurs, and bicycles. The image size is 32x32. | `https://www.cs.toronto.edu/~kriz/cifar.html` |
| CIFAR-100 | The task consists in classifying natural images (100 classes, with 500 training images each). Some examples include apples, bottles, dinosaurs, and bicycles. The image size is 32x32. | `https://www.cs.toronto.edu/~kriz/cifar.html` |
| DTD | The task consists in classifying images of textural patterns (47 classes, with 120 training images each). Some of the textures are banded, bubbly, meshed, lined, or porous. The image size ranges between 300x300 and 640x640 pixels. | (Cimpoi et al., 2014) |
| Pets | The task consists in classifying pictures of cat and dog breeds (37 classes with around 200 images each), including Persian cat, Chihuahua dog, English Setter dog, or Bengal cat. Images dimensions are typically 200 pixels or larger. | `https://www.robots.ox.ac.uk/~vgg/data/pets/` |
| Sun397 | The Sun397 task is a scenery benchmark with 397 classes and, at least, 100 images per class. Classes have a hierarchy structure, and include cathedral, staircase, shelter, river, or archipelago. The images are (colour) 200x200 pixels or larger. | `https://vision.princeton.edu/projects/2010/SUN/` |
| Flowers102 | The task consists in classifying images of flowers present in the UK (102 classes, with between 40 and 248 training images per class). Azalea, Californian Poppy, Sunflower, or Petunia are some examples. Each image dimension has at least 500 pixels. | `https://www.robots.ox.ac.uk/~vgg/data/flowers/102/` |
| SVHN | This task consists in classifying images of Google's street-view house numbers (10 classes, with more than 1000 training images each). The image size is 32x32 pixels. | `http://ufldl.stanford.edu/housenumbers/` |
| CLEVR/count | CLEVR is a visual question and answer dataset designed to evaluate algorithmic visual reasoning. We use just the images from this dataset, and create a synthetic task by setting the label equal to the number of objects in the images. | (Johnson et al., 2017) |
| CLEVR/distance | Another synthetic task we create from CLEVR consists of predicting the depth of the closest object in the image from the camera. The depths are bucketed into size bins. | (Johnson et al., 2017) |
| Retinopathy | The Diabetic Retinopathy dataset consists of image-label pairs with high-resolution retina images, and labels that indicate the presence of Diabetic Retinopahy (DR) in a 0-4 scale (No DR, Mild, Moderate, Severe, or Proliferative DR). | `https://www.kaggle.com/c/diabetic-retinopathy-detection/data` |
| birds | image dataset with photos of 200 bird species (mostly North American). | `http://www.vision.caltech.edu/visipedia/CUB-200.html` |

Table G.5: Summary of datasets used in our experiments, part II

| Dataset | Description | Reference |
|---|---|---|
| Patch Camelyon | The Patch Camelyon dataset contains 327,680 images of histopathologic scans of lymph node sections. The classification task consists in predicting the presence of metastatic tissue in given image (i.e., two classes). All images are 96x96 pixels. | (Teh & Taylor, 2019) |
| Resisc45 | The Remote Sensing Image Scene Classification (RESISC) dataset is a scene classification task from remote sensing images. There are 45 classes, containing 700 images each, including tennis court, ship, island, lake, parking lot, sparse residential, or stadium. The image size is RGB 256x256 pixels. | (Cheng et al., 2017) |
| EuroSAT | The task consists in classifying Sentinel-2 satellite images into 10 different types of land use (Residential, Industrial, River, Highway, etc). The spatial resolution corresponds to 10 meters per pixel, and the image size is 64x64 pixels. | (Helber et al., 2019) |
| dSprites/location | The dSprites dataset was originally designed to assess disentanglement properties of unsupervised learning algorithms. In particular, each image is a 2D shape where six factors are controlled: color, shape, scale, rotation, and (x,y) center coordinates. Images have 64x64 black-and-white pixels. This task consists in predicting the x (horizontal) coordinate of the object. The locations are bucketed into 16 bins | `https://github.com/deepmind/dsprites-dataset/` |
| dSprites/orientation | We create another task from dSprites consisting in predicting the orientation of each object, bucketed into 16 bins. | `https://github.com/deepmind/dsprites-dataset/https://github.com/deepmind/dsprites-dataset/` |
| SmallNORB/azimuth | The Small NORB dataset contains images of 3D-toys from 50 classes, including animals, human figures, airplanes, trucks, and cars. The image size is 640x480 pixels. In this case, we define labels depending on the azimuth (angle of horizontal deviation), in intervals of 20 degrees (18 classes). | (LeCun et al., 2004) |
| SmallNORB/elevation | Another synthetic task we create from Small NORB consists in predicting the elevation in the image. There are 9 classes, corresponding to 9 different elevations ranging from 30 to 70 degrees, in intervals of 5 degrees | (LeCun et al., 2004) |
| DMLab | The DMLab (DeepMind Lab) is a set of control environments focused on 3D navigation and puzzle-solving tasks. The Dmlab dataset contains frames observed by the agent acting in the DeepMind Lab environment, which are annotated by the distance between the agent and various objects present in the environment. The goal is to evaluate the ability of a visual model to reason about distances from the visual input in 3D environments. The Dmlab dataset consists of 360x480 color images in 6 classes. The classes are close, far, very far × positive reward, negative reward respectively. | (Beattie et al., 2016) |
| KITTI | The KITTI task consists in predicting the (binned) depth to the vehicle (car, van, or truck) in the image. There are 4 bins / classes. | (Geiger et al., 2013) |
| ColHist | Classification of textures in colorectal cancer histology. Each example is a 150 x 150 x 3 RGB image of one of 8 classes. | `https://www.tensorflow.org/datasets/catalog/colorectal_histology` |
| UC Merced | 21 class land use image dataset | `https://usdahsi.ucmerced.edudatasets/landuse.html` |
| cars | The Cars dataset contains 16,185 images of 196 classes of cars. The data is split into 8,144 training images and 8,041 testing images, where each class has been split roughly in a 50-50 split. Classes are typically at the level of Make, Model, Year, e.g. 2012 Tesla Model S or 2012 BMW M3 coupe. | `http://ai.stanford.edu/~jkrause/cars/car_dataset.html` |

# H    TRANSFER TO VTAB

In this Section, we provide additional experiments for the transfer learning scenario and use VTAB as downstream task. Figure H.32 shows the effect of controlled experiments, scaling up model size, data size and compute for transfer learning setting on VTAB dataset. Note that these experiments are based on the standard VTAB setup (Zhai et al., 2019) that uses only 1000 examples for each dataset to reflect performance of transfer learning under a reasonable labelling budget in downstream tasks. We use the same objective function for both upstream and downstream (Sigmoid cross entropy) and update all of the pre-trained parameters during fine-tuning. Table H.6 presents results of models that are pre-trained with differed head weight decays in the transfer setup on the VTAB test set. In this setup, we use SGD momentum with batch size 512 for fine-tuning all the parameters of the model using the training set of the downstream task.

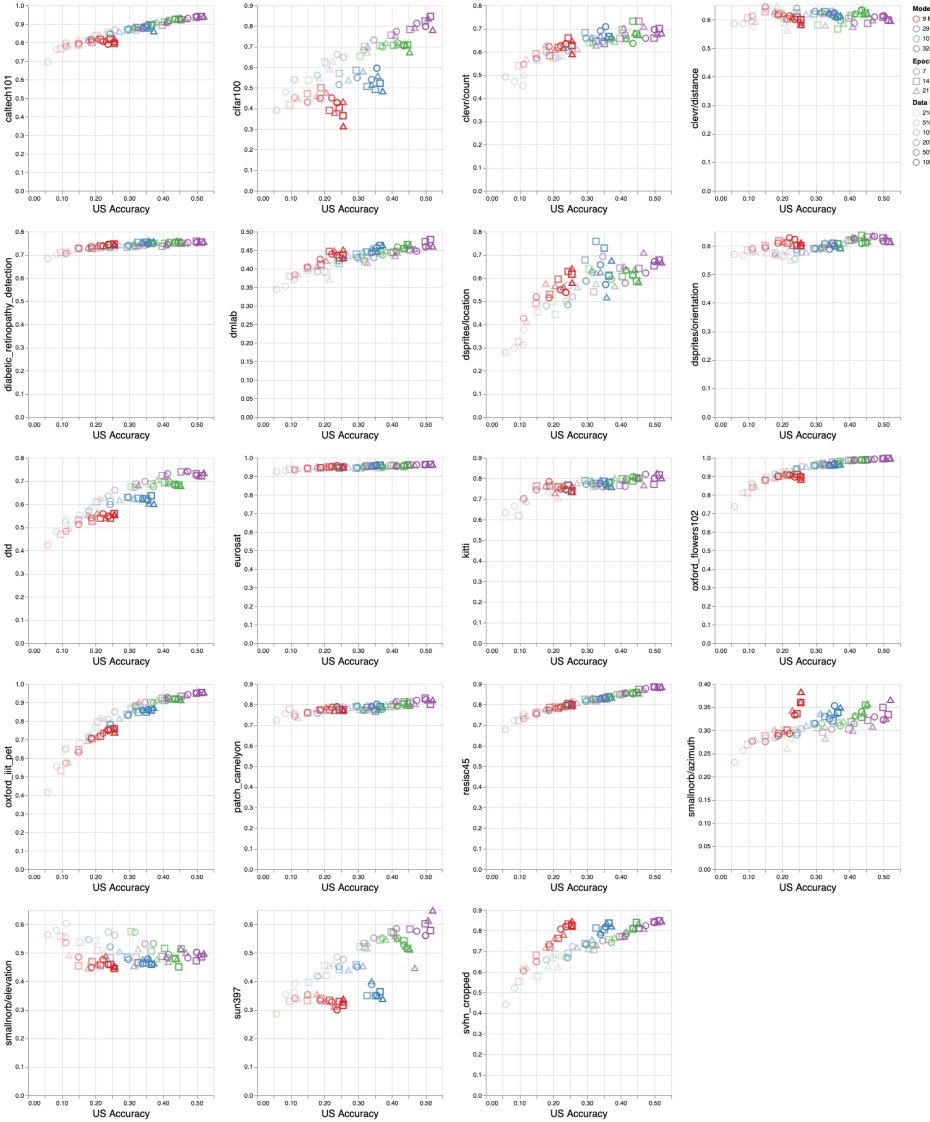

Figure H.32: Effect of controlled scale up with respect to the model size (number of parameters), data size (portion of the pre-trained data), and compute (epochs) on tasks in VTAB-1K benchmark (1000 training example per task) in the transfer setup.

Table H.6: Results of a ViT-B/32 on fine-tuning (transfer) setup on VITAB-1K benchmark, when pre-trinained with different head weight decays. Note that the selected head WD for these experiments are set to $0$ and $5.0$, which are rather extreme values, to highlight the effect on different datasets.

| Dataset | HWD=0.0 | HWD=5.0 |
|---|---|---|
| caltech101 | 0.89 | **0.91** |
| cifar100 | 0.51 | **0.79** |
| clevr-count | **0.72** | 0.42 |
| clevr-distance | **0.65** | 0.49 |
| diabetic-retinopathy-detection | **0.74** | 0.72 |
| dmlab | **0.42** | 0.36 |
| dsprites-location | **0.68** | 0.56 |
| dsprites-orientation | 0.58 | 0.58 |
| dtd | 0.66 | **0.72** |
| eurosat | 0.94 | **0.95** |
| kitti | **0.76** | 0.70 |
| oxford-flowers102 | 0.98 | **0.99** |
| oxford-iiit-pet | 0.93 | **0.94** |
| patch-camelyon | **0.78** | 0.77 |
| resisc45 | 0.82 | 0.83 |
| smallnorb-azimuth | **0.27** | 0.22 |
| smallnorb-elevation | **0.47** | 0.36 |
| sun397 | 0.42 | **0.65** |
| svhn-cropped | **0.72** | 0.60 |
| VTAB-Natural | 0.69 | **0.78** |
| VTAB-Specialized | 0.82 | 0.82 |
| VTAB-Structured | **0.57** | 0.46 |
| VTAB-ALL | **0.69** | 0.68 |

