# OpenReview forum: "Exploring the Limits of Large Scale Pre-training"
_ICLR.cc/2022/Conference — ICLR 2022 Spotlight_

### Official Review · Reviewer_3WYy · 2021-10-29

**Correctness:** 4
**Technical Novelty And Significance:** 3
**Empirical Novelty And Significance:** 3
**Recommendation:** 8
**Confidence:** 3

**Main Review:**

Strengths:
1). The relationship between (US, DS) accuracy is intuitively convincible especially in the high performance regions (Fig. 1), Appendix Table F.1, F.2 further shows that correlation values change drastically for different choices of (US, DS) tasks. Thanks author[s] for the extensive experiments.

2). The choice of hyperparameters is important for minimizing the discrepancy between US and DS performances. Appendix C and D show an case study of US weight decay in the projection layer of ViT on the effect of downstream DS accuracy, showing that at the expense of hurting US accuracy could instead lead to the improvement of DS tasks. This is counter-intuitive but technically sounds especially for high performing regions.

3). The proposal for improving performance on the breadth of downstream task by adding upstream data diversity to (data size, model parameters, FLOPs) sounds plausible. It would be excellent if author[s] can give a concrete method on how to do it in future.


Weakness:
1). If there is any case study that showing US accuracy can accurately predict DS accuracy but (model size, US data size, compute) cannot? Compared to Figure 1, which conclusion cannot not be shown from Figure 4?

2). From the study of head weight decay can we conclude that improving the feature extraction performance of lower layers lead to better generalization performance in ViT (see Figure 6, Appendix F.22, F.23)?

3). In practice it costs long time to get a single US accuracy, if there is any proxy/surrogate way to make a cheap estimation? From Figure 1 I can see the curve is saturated only towards the end, which means an early prediction of US accuracy is not a reliable predictor of DS accuracy. In other words, how can we use power law prediction for model selection in practice (see Appendix Fig. F6, F7, F8, F9)?

**Summary Of The Paper:**

This paper gives an important finding that overperformed upstream performance may not transfer well to better downstream performance especially in higher upstream accuracies region. The experiments are both empirically and theoretically evaluated, making the conclusion strong and reliable.

**Summary Of The Review:**

This paper gives an interesting study on the prediction power of US accuracy. Extensive results are provided. The conclusions are solid.

---

> ### Author Response · Authors · 2021-11-20
> **Response  to Reviewer 3WYy**
>
> Dear reviewer 3WYy,
>
> Thank you so much for the insightful review and the interesting comments and questions.
>
> Hereby, we respond to each of your concerns:
>
> 1. We don’t argue that model size, US data size, and compute can not predict DS accuracy, we argue that all these scaling variables contribute toward predicting DS accuracy, but that most of their impact is captured by the US accuracy (Figure F.18). That means, taking upstream accuracy into account, we do not need to look at these three dimensions separately. This is very useful, since modelling the scaling behaviour with respect to these three aspects, separately, is very difficult due to their complex interaction with each other and the optimal hyperparameters for each configuration.
> Figure 1, shows the big picture that, as the US accuracy increases, DS accuracy starts to saturate on many DS tasks. Where we can achieve a better US accuracy by tuning hyper-parameters or scaling up model size, data size and/or compute. Figure 4, shows the effect of each of these scaling variables in a controlled setup (it’s a grid search of all combinations of model size, datasets and training epochs). If we add enough points to Figure 4, e.g., by including more variations of hyper-parameters, we should be able to exactly replicate Figure 1.
>
> 2. Thanks for sharing this interesting insight in the review. It sounds very intuitive that generally the feature extraction mechanism can impact the generalization behaviour of the model. What is suggested as a potential conclusion from the weight decay experiment could be the case. However, this statement requires defining the type of generalization we refer to, and running more experiments and ablation studies.  What our analyses mainly indicate is that depending on the relation between the two tasks (upstream and downstream), learned features at different levels of abstraction would be transferable to the downstream task.
>
> 3. The main goal of our paper is to provide an analysis to understand the limits of scaling up. In our analysis, we estimate the best possible achievable accuracy on a given downstream task, with respect to the upstream accuracy. It’s not about estimating the downstream performance of a specific model, rather the general scaling dynamics. As an alternative approach, as we see in Figure 6, tracking the transfer performance of the representation obtained from different layers of the model, might help in predicting the scaling dynamics.
>
> We would be happy to discuss if there are any additional concerns or questions.

---

### Official Review · Reviewer_2EkD · 2021-10-29

**Correctness:** 3
**Technical Novelty And Significance:** 2
**Empirical Novelty And Significance:** 3
**Recommendation:** 8
**Confidence:** 3

**Main Review:**

Strengths:
- the high-level question that the paper asks makes sense, and becomes more important as models and datasets grow larger and larger, so that fewer organisations have the resources to handle them
- the sheer size of the study makes it interesting, I would think that many of us are doubting whether some of our findings and beliefs from data- and compute-limited studies hold also for truly huge datasets and models beyond our resources
- it is a clevert idea to analyse model performance in terms of a convex envelope, so as to better approximate the upper performance bound for a given model size and "factor out" architecture and hyper-parameter choices, and at least at first glance it makes a lot of sense

Weaknesses:
- concerns remain whether the "hyper-modern" types of models that are the focus of the paper are representative, I would argue that the majority of computer vision users still employ "old-fashioned" convolutional architectures rather than vision transformers or MLP-mixers,
especially in data- or compute-constrained scenarios
-the narrow focus on low-shot scenarios and on image-level classification is somehow inconsistent with the "real-world, large-scale" aspirations of the paper; let's face it, at the application level outside the research lab, where transfer learning and dataset constraints really matter, one-label-per-image retrieval-type settings are the exception. The effect of large-scale pre-training  for more structured tasks like
semantic segmentation, detection, etc. would be a lot more relevant
- the paper is overly dense  and overloaded. I am fine with an arbitrarily large and detailed appendix, but the paper itself tries to squeeze too many things into 9 pages. Graphs are so small they are hardly readable; explanatory text is somewhat hard to follow and parse, despite the not overly complicated message; countless references to the appendix make it practically impossible to read the main paper without the appendix; etc.

**Summary Of The Paper:**

The paper presents an empirical study (respectively meta-study) of large-scale supervised pre-training for image recognition tasks. By analysing lots of experiments with varying model sizes, dataset sizes and training durations, the paper reaches the conclusion that simply scaling them up for a generic pre-training task will not lead to proportional gains for downstream tasks that build on the pre-trained model. On the contrary, the paper suggests that with increasing pre-training effort the downstream performance will reach a saturation level below its Bayes error. Moreover, that saturation level appears to vary across different downstream tasks (i.e., image characteristics and class definitions), which is seen as a sign that a generic one-fits-all feature extractor cannot be found by just scaling up.

**Summary Of The Review:**

A brute-force, unashamedly empirical study of large-scale pre-training, albeit in a relatively specialised setting (low-shot, mostly with very recent, not yet widely established network types). The truly large size of the study means it will be interesting and potentially helpful for many people - we probably have to be grateful for such empirical studies, as very few would be able to replicate them, even if they are only "aggregated" and not "trained for the purpose of the study" (although it is perhaps a bad sign for the community that a small oligarchy of organizations now are the only ones who have that capacity). It is the sort of paper that does not strike me as particularly brilliant, but whose message will, and should, raise some eyebrows and spark discussions in the community.

---

> ### Author Response · Authors · 2021-11-20
> **Response  to Reviewer 2EkD**
>
> Dear Reviewer 2EkD,
>
> Thank you for your review and for sharing your valuable feedback with us.
>
> We certainly agree with the reviewer that one of the important aspects about this paper is the large scale meta-study. However, we would like to highlight a few of other contributions of the paper that are beyond an exhaustive study of large-scale pre-training:
> - We suggest, and provide intuitive, theoretical and empirical justification that when studying scaling behaviour, one should take the convex hull of points into account, instead of all data points or their average.
> - We show that the few-shot transfer performance of the models across their layers, correlates with the scaling relation between upstream and downstream tasks.
> - We investigate the role of hyperparameters of pre-trained models and showcase scenarios where performances of different downstream tasks, or performances of upstream and downstream tasks are at odds with each other. We also elaborate on the reason behind this phenomenon.
>
> Here we respond to the issues you have raised:
> - Concerns about studied model architecture: We’d like to restate that in our meta study we include variants of ViT and MLP-Mixers, as well as  best performing ResNet models. In fact, one of our contributions here, is to study the scaling behaviour, by looking into the accuracies on upstream and downstream tasks, irrespective of other hyper-parameters, including model architecture. The idea is to understand how much we can gain by training a perfect model on a given upstream task. This means the focus is on models that achieve best performance on the downstream task, for a fixed upstream accuracy. And we believe our study includes those. Moreover, there is evidence indicating that choice of architecture does not impact the power law governing the DS performance (see [2,3,4,5] for example).
>
> - Concerns about practicality of few shot settings: In our analysis we investigate few-shot transfer performance of the models where the number of shots is 1,5, 10, and 20. We observe similar trends for different numbers of shots. Moreover, we have included fine-tuning experiments as proof of concept, to show that our conclusions generalize to this setup as well. While in many cases, the few shot performance of the models is relatively a good indicator of fine-tuning performance [1], few-shot settings, itself,  is one of the most commonly used setups in practical scenarios.
>
> - Tasks byond classification: We agree that extending these analyses beyond image classification is an important next step. However, in this paper we focus on image classification to ensure the training objective is similar for upstream and downstream and that there are no side effects due to task discrepancy. We are very eager to expand our study and are planning to go not only beyond classification, but also beyond vision and explore more diverse setups. These however require dedicated studies. The main challenge for this goal is collecting enough experiments both in terms of quantity and diversity.
>
>
> ### References:
> 1. Djolonga, Josip, et al. "On robustness and transferability of convolutional neural networks." Proceedings of the IEEE/CVF Conference on Computer Vision and Pattern Recognition. 2021.
> 2. Kaplan, Jared, et al. "Scaling laws for neural language models." arXiv preprint arXiv:2001.08361 (2020).
> 3. Shankar, Vaishaal, et al. "Do Image Classifiers Generalize Across Time?." Proceedings of the IEEE/CVF International Conference on Computer Vision. 2021.
> 4. Shankar, Vaishaal, et al. "Evaluating machine accuracy on imagenet." International Conference on Machine Learning. PMLR, 2020.
> 5. Taori, Rohan, et al. "Measuring robustness to natural distribution shifts in image classification." arXiv preprint arXiv:2007.00644 (2020).

---

### Official Review · Reviewer_SKn8 · 2021-11-01

**Correctness:** 4
**Technical Novelty And Significance:** 3
**Empirical Novelty And Significance:** 3
**Recommendation:** 6
**Confidence:** 3

**Main Review:**

Strengths:

1. The paper is well-written and easy to follow.

2. Extensive experiments and analysis have been conducted, which makes it a strong technical report.

3. Besides the saturating behavior, some insights drawn from the observations are good to be heard in the community, such as increasing the data density of upstream tasks, predicting downstream performance for a given upstream accuracy, and etc.

Weaknesses:

1. This paper is just focusing on the task of image recognition. It will be more beneficial to consider complicated visual tasks and other modalities and to investigate whether the same conclusion still holds.

2. In terms of predicting downstream performance for a given upstream accuracy, can the authors compare the proposed method with those model selection techniques such as LogME [1]? Which technique is more practical for the real-world transfer tasks to select the best pre-trained model?

[1] You, Kaichao, et al. "Logme: Practical assessment of pre-trained models for transfer learning." International Conference on Machine Learning. PMLR, 2021.

**Summary Of The Paper:**

This paper conducts extensive experiments on ViTs, MLP-Mixers and ResNets with up tp 10B parameters and evaluates transferability on more than 20 downstream tasks. It draws the conclusion that as we improve the performance of the upstream task, the performance of downstream tasks shows a saturating behavior.

**Summary Of The Review:**

Overall, the reviewer tends to vote for accept since this work makes huge efforts in terms of experiments and some of the conclusions are beneficial to be heard in the community.

---

> ### Author Response · Authors · 2021-11-20
> **Response to Reviewer SKn8**
>
> Dear reviewer SKn8,
>
> Thank you for your comments. We appreciate that you find the paper well written and some of the conclusions beneficial for the community.
>
> We definitely agree with the reviewer that it is, indeed, important to run such analyses for different types of tasks and modalities.  One concern that we take into account while designing our experiments is to avoid side effects from uncontrolled factors. An example of this is when upstream and downstream tasks are different. E.g., classification (recognition) vs object detection(localization). In scenarios like this, task discrepancy could make the learned knowledge from pre-training in general less transferable to the downstream tasks [1].  The phenomena that is under study in our paper is by itself complex. Hence, in this work, we decided to focus on a setup where the task/objective is similar in upstream and downstream, which is image recognition. Moreover, many of the analyses we ran are based on the results from experiments that were already done by researchers. At least currently, we don’t have access to such a large number of experiments for other pre-training tasks than image recognition to be able to deliver robust conclusions.
>
> It is both interesting, and a curious case, to study the scaling behaviour of the models in various setups including when we have different objectives and tasks in upstream and downstream, but it’s only fair if we postpone these analyses to future work.
> About comparison with the recent work by You et al 2021, per request of the reviewer, we have added a paragraph to the related work section, to compare what we do in this paper with methods for estimating transferability of pre-trained models, including LogMe.
> Here is what we have added to the “additional related work” section of the paper (Appendix A)
> >In the context of transfer learning, generally, model selection is a major step. Given a set of pre-trained models,  and a downstream task, we need techniques or methods to help us predict the performance of the pre-trained models on the downstream task. There have been several efforts in this direction, including NCE[3], LEAPS[4],  and most recently LogME[2]. The main focus of the aforementioned methods is to estimate the transfer performance of the models efficiently without actually having to fine-tune them on the given downstream task.
> In this paper, on the other hand, our aim is not to predict the transfer performance of a specific model. We are taking the big picture into account, and focusing on understanding the limits of scaling up, i.e., the dynamics of the scaling behaviour of DS performance with respect to US accuracy.  The framework we propose and employ in this paper, can in general be used to answer the question of “How beneficial it would be for the downstream task if we improve the performance of the models on the upstream task by scaling up data, compute, model-size?”.
> We show that having enough data points with a wide enough range of accuracies on the upstream task, one can fit a power-law curve that can be used to predict the best possible accuracy on the DS task with respect to any given us accuracy. Ideally, this would mean that we can predict the performance of models that do not yet exist. (We are predicting the best performance one could achieve on a DS task by pretraining on a given US  task).
> Here we use few-shot accuracy as an indicator of the performance of the models on the downstream tasks. Methods such as LogME can be beneficial for such analysis, e.g., instead of tracking the few-shot performance of the models, one could look into other transferability metrics.
>
> Finally, we sincerely hope that we have addressed all your concerns in the rebuttal. We would be happy to discuss if there are any further points. Considering that you find the technical and empirical novelty of our paper significant, and all the statements to be correct and well supported, we appreciate it a lot if you can update your overall assessment score of the paper if there are no additional issues.
>
> ### References
> 1. Zoph, Barret, et al. "Rethinking pre-training and self-training." 34th Conference on Neural Information Processing Systems (NeurIPS 2020).
> 2. You, Kaichao, et al. "LogME: Practical assessment of pre-trained models for transfer learning." International Conference on Machine Learning. PMLR, 2021.
> 3. Tran, A. T., Nguyen, C. V., and Hassner, T. "Transferability and hardness of supervised classification tasks." In ICCV, 2019
> 4. Nguyen, C., Hassner, T., Seeger, M., and Archambeau, C. "LEEP: A New Measure to Evaluate Transferability of Learned Representations." In ICML, 2020.

---

### Official Review · Reviewer_pqjL · 2021-11-04

**Correctness:** 4
**Technical Novelty And Significance:** 2
**Empirical Novelty And Significance:** 4
**Recommendation:** 8
**Confidence:** 3

**Main Review:**

### Key Strengths
  1. This problem is very important, and these anlyses targeted to assess a key part of that problem.
  2. The scale of these experiments is impressive, and their findings robustly motivated by said scale. It is hard to argue with the observed saturation effect in the context of how extensively you've profiled it. Similarly, the results you've found are I think impactful. You've clearly established that we _cannot_ use upstream accuracy as proxy for _downstream accuracy_, and quantified some other insights as well relating to task consistency. These should be impactful in future research, though some revisions could amplify that effect.
  3. I think that this is a meta-analysis makes the work much stronger--one of my biggest concern at first skim of the paper was precisely that your models wouldn't be trained in a manner consistent with other models published during ordinary research, and thus that your results would be less generalizable. By largely relying on pre-published models and results, you totally alleviate that concern and strengthen your findings.
  4. While I think you could do more here, I really like the representational analysis as well. Showing the consistency between saturation speed and layer preference both establishes some intuitive precedent for your findings and helps hint at some strategies we could take in the future to improve these results as well.

### Key Weaknesses
  1. Your biggest issue here is presentation. This presents itself in two ways. Firstly, you are _constantly_ referring to the appendix. If those appendix plots are so essential to your work, you need to find a way to incorporate them into the main body, or look for a different venue without page limitations (e.g., JMLR). Secondly, the flow of this paper somewhat hinders my ability to get to your key takeaways quickly. I like your introduction -- I think it does a good job of setting up the problem and framing what you're going to do, but I feel like you lose that framing as soon as you branch into other sections. I sort of find myself reading the paper by skipping from figure to figure and trying to back out what take-aways the figures represent from the text, when what you want is to guide the reader through your work with your text directly. To address this, I'd recommend reformulating the work slightly to have a more clear-cut structure, and generously sprinkle forwards and backwards pointers throughout your text and contextulaization sections to keep the reader constantly aware of where they are in your analysis and how that contributes to the big picture. You may also need to relegate some of your findings out of the main text, to secondary publications, or drop them entirely if you really want to crystallize your main take-aways for your readers.
  2. Your work already speaks to the fact that for some tasks you can establish a strong power law fit even without using the full dense dataset you use in your experiments. However, in order for this take-away to be proactively useful in general, it would also be valuable to determine to what extent you can fit a power-law saturation curve to a new model with only a limited computational budget for pre-trained models -- e.g., how few runs and over how few UA data-points can you accurately assess the power-law fit? Additional analyses on these fronts (or clear callouts/explanations to your current findings on these fronts) would be very valuable.

### Minor Weaknesses
  1. I think you probably spend too much time on the randomized classifier section. I don't think those points are central to your paper, and I think this content could therefore be largely relegated to an appendix.
  2. I think using accuracy is standard in the field, but I'd be further interested in other metrics like AUROC for these analyses too.
  3. You should re-label your legend entry in Figure 1 for the asymptotic (@ UA = 1) performance value. I didn't find it clear from the legend entry alone and your caption for that figure is already very long.


### Other comments
  1. I think it'd be really impactful if you replicated this analysis on NLP models.


**Summary Of The Paper:**

### What is the Problem / Question?
There is a ongoing trend in ML right now of exploring larger and larger pre-trained (PT) models. In general, increasing the scale of these PT models has yielded better performance on downstream tasks, though this is not universally the case. This work analyzes the limits of this trend in the computer vision space, investigating how downstream task performance is impacted by scale of the PT model among other factors.

### Why is it impactful?
Pre-training large models is very expensive. This study has the potential to clarify when such large-scale PT is well-motivated, and when it isn't, which could yield significant benefits for future research.

### Why is it hard? Why have previous approaches failed?
Previous approaches have examined this question at a dramatically smaller scale than this study does. For example, Kornblith et al., 2019 examined on the order of 10s of models, as opposed to thousands of models as examined here, and the results found here are correspondingly more detailed than those found in the Kornblith et al. study. _Most critically, the authors here present data that strongly suggests that rather than a linear relationship between downstream and upstream task performance, we should instead expect a saturating, non-linear trend_, which suggests that continually training larger and larger models may not be as worthwhile as was traditionally assumed. The fact that this counters an established result in the literature is an important distinction, because my native assumption would've actually been consistent with the author's finding, so without the literature bias already existing, this finding would in my opinion be less impactful. However, because it is updating a held belief in the published results, it is much more important.

### How do they solve / answer it?
The authors profile a variety of models across different architectural choices, hyperparameter settings, etc. and compare upstream vs. downstream accuracy across various downstream tasks. They further examine a variety of additional angles within this study, including:
  1) Quantifying the extent to which downstream performance is consistent across downstream tasks at a given level of upstream task performance,
  2) How the internal representational space of these models can inform the extent of the saturation of the downstream task.
  3) They examine in which cases upstream and downstream task performance may be at odds with one another
  4) They profile their analyses under various dataset sizes, few-shot settings, and architectures.


**Summary Of The Review:**

While I think addressing both key weaknesses would greatly strengthen the work, I think this is already an impactful and highly rigorous, technically sophisticated study which meets the bar for this venue, so I vote accept here.

---

> ### Author Response · Authors · 2021-11-20
> **Response to Reviewer pqjL**
>
> Dear Reviewer pqjL,
>
> Thank you very much for your thoughtful comments and the substantive review. We appreciate that you find this paper impactful and rigorous.
>
> We also thank you for sharing your concerns about the presentation of the paper. We agree that the paper is a bit too dense. We have put a lot of thought into organizing the content of the paper, so that all the necessary information and pointers are included and clearly explained. We will make sure to work on this further for the camera ready version  to have a more concise thread of take home messages in all sections. If you have any additional concrete suggestions about how we can improve the readability of the paper, we appreciate it and will try to incorporate that in the paper.
>
> As you mentioned, in section 2 (Figure 3), we show that if we fit the power-law equation to the convex hull of the points (versus actual data points, or their average), we can estimate the scaling behaviour (downstream accuracy with respect to upstream accuracy) more accurately. While adding more data points improves the prediction accuracy (accuracy of predicting downstream performance given the upstream performance), with as few data points of 500 we can achieve a relatively low prediction error. The main point here is that the data points should cover a wide range of US accuracies, otherwise, only taking a small range of US accuracies and their corresponding DS accuracies into account could lead to miscapturing the scaling behaviour to be linear.
>
> On the other hand, If the question is:  “How much scaling up to get a better performance on the upstream task can contribute toward achieving a better performance on the downstream task?”, our analysis, in Figure 6, indicates that the performance of the downstream task on subnetworks of the model (i.e., when moving head to lower layers of the model), can be a good qualitative indicator.  So, given a new pair of DS and US tasks,  in the minimal case, even one model could show the scaling relationship between the two tasks.
>
> We would like to emphasize that one of the contributions of our paper is proposing the use of the convex hull of the data points when studying the scaling behaviour. While our focus is on the relation between upstream and downstream accuracy, this technique should generally be applied in most cases where scaling laws are being investigated to reduce the effect of noise and remove the dependency on density of the points in the plots.  The discussion about randomized classifiers is to justify why we can rely on the convex-hull of the points to get a more accurate fit for the scaling function. The point is that we are taking into account the performance of models that don’t exist in our dataset, and the argument of the randomized classifier provides  a proof that there definitely exists a model with such accuracies.
>
> We agree that it is interesting to study the scaling laws considering a variety of metrics, as the performance of the model with respect to different metrics is not always correlated. But considering the fact that the paper is already too dense, as rightfully mentioned by the reviewer, we believe it can be considered sufficient for now to report the results based on the commonly used metric, accuracy, and postpone further analysis with a wider range of performance indicators to future work.
>
> Finally, we couldn’t agree more with you on the point that it would be both very interesting and beneficial to study the same phenomena for NLP tasks. In fact, we already mentioned this  as an important next step in future work. However, there are two main challenges for this future direction: (1) Gathering data for empirical study of this scale is not so trivial, and at least for now we don’t have access to such data for NLP models (we would need models all trained on the same upstream task). (2) Moving to language modality, we will face new challenges. For instance,  the fact that the most common setup is to use self-supervised pre-training for upstream  (e.g., language modeling objective) and supervised fine tuning on various tasks. Discrepancy between the US/DS objectives would introduce a new factor that requires special treatment when running analyses.
>
> We also updated the legend of Figure 1, hopefully it is a bit more clear now.

---

### Decision · Program_Chairs · 2022-01-20

**Decision:**

Accept (Spotlight)

**Comment:**

This paper provides a very large-scale study on the pretraining of image recognition models. Specifically, three scaling factors (model sizes, dataset sizes, and training time) are extensively investigated. One important phenomenon observed by this paper is that stronger upstream accuracy may not necessarily contribute to stronger performance on downstream tasks---actually sometimes these two types of performance could even be at odds with each other

Overall, all the reviewers enjoy reading this paper and highly appreciate the empirical results presented in this paper. There were only a few concerns raised by the reviewers but most were well addressed during the discussion period. All reviewers reach a consensus on accepting this paper and believe this study is worthy to be heard by the community.